# A cross-scale study for compound flooding processes during Hurricane Florence

Fei Ye[1], Wei Huang[1], Yinglong J. Zhang[1], Saeed Moghimi[2], Edward Myers[2], Shachak Pe'eri[2], Hao-Cheng Yu[1]

[1]Virginia Institute of Marine Science, College of William & Mary, Gloucester Point, 23062, USA

[2] NOAA National Ocean Service, Silver Spring, 20910, USA

*Correspondence to*: Fei Ye (feiye@vims.edu)

## Abstract

We study the compound flooding processes as occurred in Hurricane Florence (2018) that was accompanied by heavy precipitation, using a 3D creek-to-ocean hydrodynamic model. We examine the important role played by barrier islands in

the observed compound surges in the coastal watershed. Locally very high resolution is used in some watershed areas in order to resolve small features that turn out to be critical for capturing the observed High Water Marks locally. The wave effects are found to be significant near barrier islands and have contributed to some observed over-toppings and breaches. Results from sensitivity tests applying each of the three major forcing factors (oceanic, fluvial, and pluvial) *separately* are succinctly summarized in a "dominance map" that highlights significant compound effects in most of the affected coastal

watersheds, estuaries and back bays behind the barrier islands. Operational forecasts based on the current model are being set up at NOAA to help coastal resource and emergency managers with disaster planning and mitigation effort.

## 1 Introduction

Recently, more frequent occurrences of "wet" hurricanes (i.e., hurricanes accompanied by heavy precipitation) that stall

near the coast (Pfahl et al., 2017; Hall and Kossin, 2019) have brought new challenges to coastal communities in the form of compound flooding, which is defined as concurrence of flooding from same or different origins (river, storm surge and rainfall), especially in the coastal transitional zone that sits at the border between coastal, estuarine, and hydrologic regimes (Santiago-Collazo et al., 2019). Compound flooding highlights one of the major pitfalls of the current hurricane intensity scale, which is entirely based on wind speed, leaving the potential rainfall and flooding impacts to be glossed over in initial

forecasts that emphasize hurricane category. The record-setting 2020 Atlantic hurricane season (which has several very wet storms) highlights the urgency and exposes the current knowledge gap for understanding compound flooding processes.

A recent example for compound flood events is Hurricane Florence that impacted a large area of North Carolina (NC) in September 2018. Hurricane Florence was the first major hurricane of the 2018 Atlantic hurricane season. Originated from a

strong tropical wave near Cape Verde, west Africa, it acquired tropical storm strength on September 1, followed by a rapid intensification to a Category 4 status on September 4, with estimated maximum sustained winds of 130 mph, and eventually reached its maximum strength on September 11. It made landfall south of Wrightsville Beach, near the border between NC and South Carolina (SC) as a Category 1 hurricane on September 14. The slow motion of the storm after the landfall brought heavy rainfall throughout the Carolinas for several days. Compounded by the storm surge, the rainfall caused widespread flooding along a large swath of the NC coast, and inland flooding in cities such as Fayetteville, Smithfield, Lumberton, Durham, and Chapel Hill. According to a USGS report (Fester et al., 2018), a new record rainfall total of 35.93 inches was set during the hurricane in Elizabethtown, NC. Many other locations throughout NC and SC also set new rainfall records (Fig. 1). Florence is a quintessential example of major flooding caused by a slow moving, moisture-laden storm, even if it does not have strong hurricane wind.

In this paper, we will study the response to the storm in the watershed rivers and estuaries and examine the processes and sources that lead to the compound flooding there. We will also examine the coastal responses to the event and the close connection between watershed and coastal ocean. The existing modeling efforts on compound flooding (Chen et al., 2010; Cho et al., 2012; Dresback et al., 2013; Chen and Liu, 2014; Ikeuchi et al., 2017; Kumbier et al., 2018; Pasquier et al., 2019; Wing et al., 2019; Muñoz et al., 2020) often focus on a subset of the processes (storm surges, tides, waves, fluvial flooding, pluvial flooding, and potential baroclinic effects), leaving gaps in accurately representing the complex interactions among them (Santiago-Collazo et al., 2019). What distinguishes this study from traditional compound flooding simulations is a holistic approach that solves interrelated processes in different regimes and on multiple temporal/spatial scales with the same hydrodynamic core (i.e., the same set of governing equations) in a single modeling framework. An overview of the processes studied in this paper is shown in Fig. 2. The primary tool used in this study is a proven cross-scale 3D baroclinic model designed for effective and holistic simulation of intertwined processes as found during this event. The trade-off between 2D and 3D setups was carefully weighed with the goal of operationalization in mind before the 3D set up was chosen. In short, the advantage of 2D is the speed (about 80 times faster than its 3D baroclinic counterpart) and the simplicity of the set up; the disadvantage is that it misses baroclinic effects and 3D processes whose importance can vary in space and time. For example, the baroclinic effects during the adjustment phase after Hurricane Irene (2011) are discussed in detail in Ye et al. (2020), using a similar model setup as the one used here. Even though different setups (2D, 3D barotropic, and 3D baroclinic) were tuned to their best possible skills, the 3D baroclinic setup was shown to better capture the total elevation during the post-storm adjustment phase. In addition, during the ongoing effort to operationalize the model, we found that including 3D processes greatly simplified the bottom friction parameterization at some coastal locations (e.g., NOAA Station 8447930 at Woods Hole, MA; Huang et al., submitted). A 3D model can also produce relevant 3D variables (e.g., 3D velocity and tracer concentration) that are important for safe navigation and ecosystem health. The 3D model presented in this paper is efficient enough for operational forecasts (see Section 3.1), which are being set up at NOAA (National Oceanic and Atmospheric Administration).

The rest of the paper is structured as follows. Section 2 will review the study site and available observations collected during the event in the watershed, estuaries, and coastal ocean. Section 3 describes the numerical model used and its set-up. Section 4 presents model validation and important sensitivity test results; the validation is done in a cross-scale fashion from small-scale watershed areas to large-scale coastal ocean. Section 5 discusses the compound effects as revealed by the 3D model in all regimes. Section 6 summarizes the major findings and planned follow-up work.

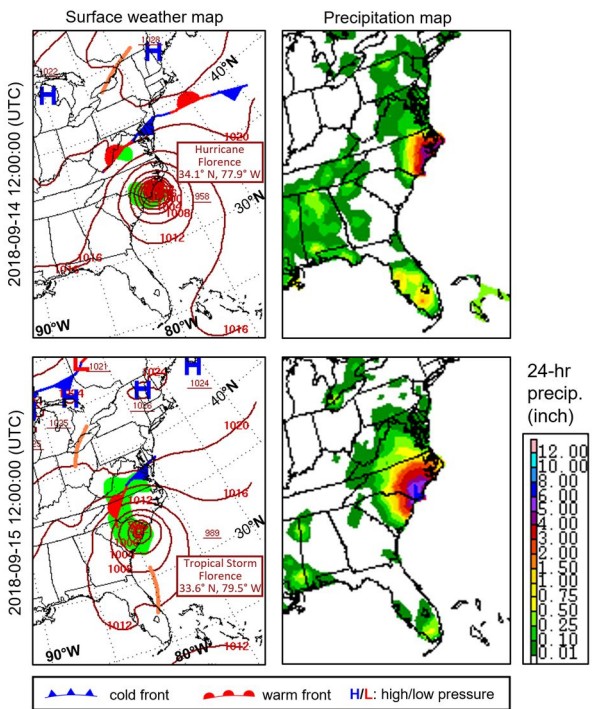

**Fig. 1: Weather map showing the low-pressure system and amount of rainfall Florence brought to North Carolina and South Carolina coast around landfall. Credit: NOAA Central Library U.S. Daily Weather Maps Project (https://www.wpc.ncep.noaa.gov/dailywxmap/); partial views of the original online maps.**

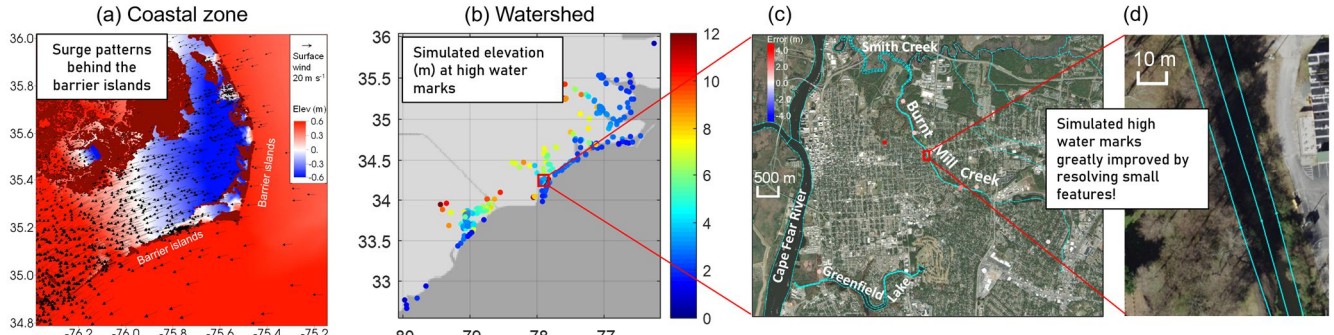

**Fig. 2: Overview of the processes studied in the paper, from (a) coastal zone to (b) watershed, and down to very small local scales in the watershed in (c) and (d). The base maps in (c) and (d) are provided by ESRI.**

## 2 Study site and observation

The focus (high-resolution) area of this study is the NC and SC coast and coastal watersheds that saw most of the impact from Florence (Fig. 3b and Fig. 3f). Similar to what we did for other hurricane events, the spatial domain's landward boundary is set at 10 m above the NAVD88 datum, which is deemed sufficient to capture most backwater effects (Zhang et al., 2020). A rich set of observations for physical and biological variables are available from satellites, autonomous instruments (e.g., gliders and Argo floats), in situ stations operated by NOAA and USGS's field estimates collected during after-event surveys (e.g., High Water Marks or "HWMs"; Fig. 2b and Fig. 2c). Analysis and quality control of these datasets have been done by the data distributors, together with uncertainty assessments. Some of the datasets will be presented in the context of model validation sections below to allow for a comprehensive and objective assessment of the model errors and uncertainties. Assessment of compound flood models such as ours inevitably involves observation collected at disparate spatial and temporal scales of several orders of magnitudes contrasts, as illustrated in Fig. 2. To the best of our knowledge, this type of model assessment has rarely been attempted before even in a 2D setting, due to the formidable challenges to numerical models (Santiago-Collazo et al., 2019), but is badly needed in order to gain a holistic understanding of the complex processes at play (Ye et al., 2020; Zhang et al., 2020; Huang et al., 2021).

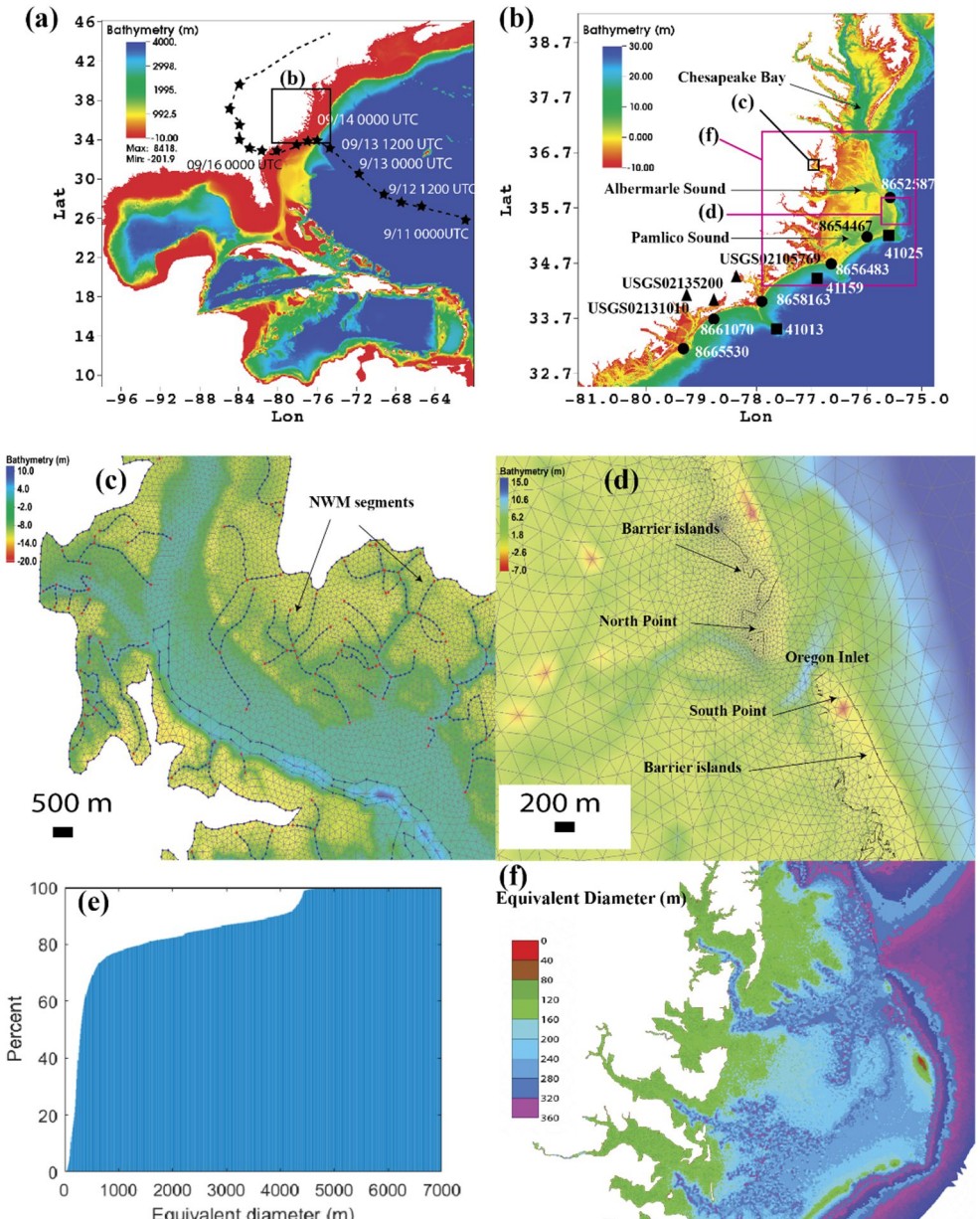

Fig. 3: Model domain and horizontal grid. (a) Domain extent and hurricane track. (b) Station locations along the North Carolina and South Carolina coast. The six NOAA gauges are: Charleston (8665530); Springmaid Pier (8661070); Wrightsville Beach (8658163); Beaufort (8656483); Hatteras (8654467); Oregon Inlet (8652587). The three squares are NDBC buoys (41013, 41159, 41025). The spatial extents of (c), (d), and (f) are also marked in (b). (c) Zoom-in of grid in a watershed area (the arcs are from NWM river network). (d) Zoom-in of grid near barrier islands and inlet (the dark line is the 0 m isobath, NAVD88); (e) Cumulative histogram of grid resolution (measured in equivalent diameters); (f) Grid resolution in North Carolina's coastal watershed area.

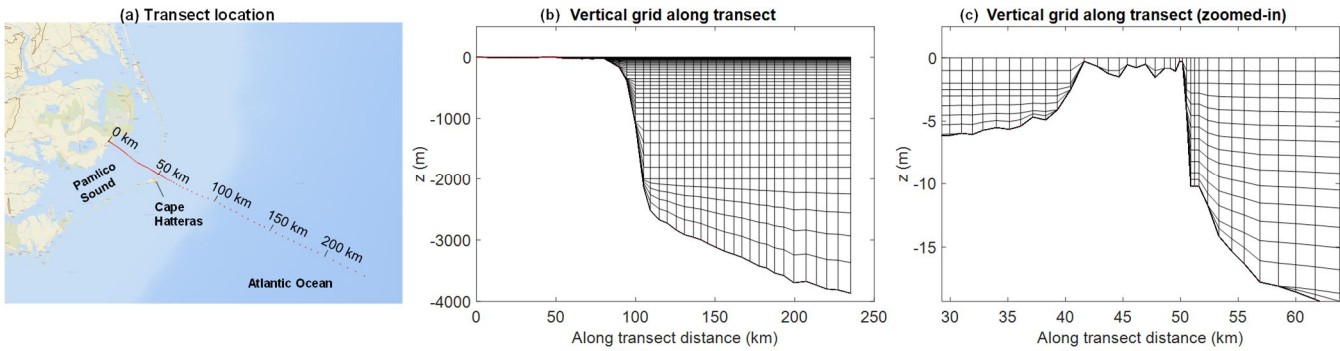

**Fig. 4: Vertical grid. (a) Transect from the watershed to the ocean, used to illustrate the vertical grid; (b) vertical grid along the transect; (c) zoom-in from (b), illustrating the transitions from 3D (Pamlico Sound) to 2DH (barrier island) and back to 3D (coastal ocean). The base map in (a) is provided by ESRI.**

## 3 Model description

### 3.1 Model setups

To capture the storm surge effects we use a large study domain that encompasses the north Atlantic west of 60°W (Fig. 3a). An added benefit of using such a large domain in conjunction of a 3D baroclinic model is that the interaction between large- and small-scale processes can be organically examined in a single model. For example, the disruption and oscillation of Gulf Stream by storms can directly affect the coastal inundation (i.e., the fair-weather flooding reported by Ezer (2018)); our results suggest that the converse is also true, as watershed processes can also affect Gulf Stream and other coastal processes (Ye et al., 2020). Therefore, a seamless creek-to-ocean model is advantageous for compound flood studies.

As we shall see, this model solves the physical processes from the watershed to the ocean with the same set of governing equations, qualifying for Santiago-Collazo et al. (2019)'s definition of a fully-coupled compound surge and flood model. SCHISM (schism.wiki) uses efficient semi-implicit solvers to solve the hydrostatic form of the Reynolds-averaged Navier-Stokes equations and transport equation (Zhang et al., 2016), which govern all flow movements inside the 3D model domain, including overland flow in the watersheds as well as estuarine and ocean circulations. Major characteristics of the model that ensure a balance of accuracy, efficiency, robustness and flexibility include: hybrid finite-element/finite-volume methods and a highly flexible 3D gridding system ("polymorphism") that combines a hybrid triangular-quadrangular unstructured grid in the horizontal dimension; and localized sigma coordinates with shaved cells (dubbed as LSC$^2$; Zhang et al., 2015) in the vertical dimension. The polymorphism allows a single SCHISM grid to seamlessly morph between full 3D, 2DH (2D depth-averaged), 2DV (2D laterally averaged), and quasi-1D configurations. The employment of shaved cells near the bottom in particular faithfully preserves the original bathymetry without any smoothing required. The detrimental effects of bathymetry smoothing on important physical and biological processes (e.g., residual transport, lateral circulation, nutrient cycling, etc.) have been documented in Ye et al. (2018) and Cai et al. (2020).

Similar to a recent compound flooding study using SCHISM for Hurricane Harvey (Huang et al., 2021), the current model domain covers the entire US East Coast and the entire Gulf of Mexico, with all major bays/estuaries and coastal watersheds resolved (Fig. 3). The horizontal grid, generated using the software SMS (aquaveo.com), has 2.2 million nodes and 4.4 million elements (Fig. 3). About 50% and 40% of the elements have resolution finer than 300 m and 220 m respectively (Fig. 3e). The grid bathymetry is interpolated from a combination of DEM sources from coarse (ETOPO1, 90-m Coastal Relief Model[1]) to fine resolution (1/9 arc-second CUDEM[2] and 1−3 m CoNED[3]), and the vertical datum used is NAVD88, with appropriate conversion between datums done by the VDatum tool (vdatum.noaa.gov). Note that NAVD88 is a more convenient datum to use in the model as most of recent observation data refer to this datum, and therefore we use this datum in the model setup and allow the model to automatically set up the sub-tidal surface slope from coastal ocean into watershed (due to the friction effects). Altogether close to 400 DEM tiles are used to cover such a large region. The horizontal grid resolution ranges from 6−7 km in the open ocean to ~400 m near the shoreline, with barrier island and narrow inlets resolved; river channels and creeks have about 300 m along-channel resolution and variable cross-channel resolutions to ensure adequate representation of the channelized flow. Shipping channels are represented by quadrangles and have 20 m or finer cross-channel resolution in NC. The finest grid resolution used is ~1 m used to represent many levees in other parts of the coast; note that as an implicit model SCHISM is not constraint by Courant–Friedrichs–Lewy (CFL) condition and thus can handle high resolution efficiently. Moreover, to better capture the geometry/bathymetry of flood pathways in the watershed region, specifically the region between the 10-m contour (set as the land boundary) and the 0-m contour of the DEM (based on NAVD88), about 300K National Water Model (NWM) segments (i.e., thalwegs; Fig. 3c) are reproduced in SCHISM's horizontal grid. Note that only the geometry of the NWM segments is retained, while NWM outputs are only used as the land boundary condition and NWM does not solve any hydrodynamics inside the model domain. Customary of all SCHISM applications, no manipulation or smoothing of bathymetry was done in the computational grid after interpolation of the depths from DEMs (including steep slopes in the Caribbean and all shipping channels). From our experience, CUDEM may underestimate the depth of coastal streams (e.g., those in the South Carolina watersheds), which is a potential error source of our model. In the vertical dimension we use 1−43 grid layers, with 43 layers being applied in the deep ocean and 1 layer in most of the watershed (Fig. 4), thus effectively rendering the model 2DH there, which is sufficient for processes like overland flow and inundation.

Table 1 shows the setups for "baseline" and important sensitivity simulations used in this paper. For the baseline, imposed at the ocean boundary (60°W) are tidal elevation and barotropic velocity of 8 tidal constituents (S2, M2, N2, K2, K1, P1, O1, and Q1), extracted from the FES2014 database[4]. The baroclinic components for the elevation and velocity are derived from

[1] https://ngdc.noaa.gov/mgg/coastal/crm.html (last accessed in Jan 2021)
[2] https://www.ncei.noaa.gov/metadata/geoportal/rest/metadata/item/gov.noaa.ngdc.mgg.dem:999919/html (last accessed in Jan 2021)
[3] https://www.usgs.gov/core-science-systems/eros/coned (last accessed in Jan 2021)
[4] https://datastore.cls.fr/catalogues/fes2014-tide-model/ (last accessed in Jan 2021)

the daily outputs of the "HYbrid Coordinate Ocean Model" (HYCOM; hycom.org). The initial condition for the water elevation is set to be 0 in all areas with positive grid depths ("wet" with 0 initial water level, e.g., rivers and bays) and to be equal to the bottom elevation in areas with negative grid depths ("dry" with 0 initial water depth, e.g., high ground in the watershed). The initial condition for the horizonal velocity is zero in the watershed and is interpolated from HYCOM elsewhere. Commensurate with the non-zero velocity (fully dynamic state) are salinity and temperature values interpolated from HYCOM; however, in the nearshore areas where HYCOM results are less accurate, the initial conditions for salinity and temperature are interpolated from the sparse observation at several USGS gauges in order to speed up the dynamic adjustment process. The surface meteorological forcing applied in the model is a combination of two products: (1) a high-resolution ERA re-analysis product from the European Centre for Medium-range Weather Forecast (ECMWF) with a ~9 km horizontal resolution (see Acknowledgements), and (2) a 3-h time interval with NOAA's High-Resolution Rapid Refresh (HRRR[5]), which is a cloud-resolving and convection-allowing atmospheric model with a 3-km horizontal resolution and a 1-h time interval. The friction of the baseline model was tuned in the wet area (river, estuary, ocean; lower than 1 m, NAVD88) and on higher grounds (higher than 3 m, NAVD88) separately. In the wet area, drag coefficients within a range of 0.001-0.01 (non-dimensional) were tested. Commonly accepted default value 0.0025 gave good error statistics near the landfall site and values within a range of 0.001-0.005 gave very similar results. In the watershed, drag coefficients within a range of 0.01-0.5 were tested. The optimal value was chosen based on the High Water Marks (HWMs) comparisons (Section 4.3) at 276 locations recorded by USGS. A small friction value within this range tended to under-predict the elevation at HWMs, and a large value led to over-prediction. Values within a range of 0.02-0.05 gave good error statistics. We chose 0.025 because it gave slightly better results in the Cape Fear River watershed near the landfall site. To sum up, drag coefficient is set at a constant value of 0.0025 at all "wet" locations, then linearly increases to 0.025 as the ground elevation increases from 1 m to 3 m (NAVD88), then a constant value of 0.025 is used for higher grounds where the bed texture is generally rougher than the riverbed. Note that this is the parameterization based on the region influenced by Hurricane Florence. Spatially varying parameterization of bottom friction for different systems is an on-going effort as we study more recent hurricanes and operationalize the model along the East Coast and Gulf Coast. However, as presented in Ye et al. (2020), Zhang et al. (2020) and Huang et al. (2021), the choices described above seem to work fine in general for other systems as well. The method used to impose the river flow in the model is described in the next subsection.

Choices of the "baseline" model parameters are similar to those used for Irene (Ye et al., 2020). The time step is 150 seconds (sensitivity tests using 100−150 s gave very similar results). The level-2.5 equation turbulence closure scheme chosen is from the generic length scale model $k$-$kl$ (Umlauf and Burchard, 2003). The simulation starts from 2018-08-24 00:00 (UTC) and lasts for 36 days to cover the hurricane and ensuing restoration period. Although the model covers a large domain, most of the elements (those in the watersheds) are quasi-2D, making it efficient enough for operational forecast. For

---

[5] https://rapidrefresh.noaa.gov/hrrr/ (last accessed in Jan 2021)

the baseline run, the real time to simulation time ratio is 80 with 1440 cores on TACC's Stampede2 cluster and 30 with 480 cores on W&M's SciClone cluster. Intel Skylake cores with a nominal clock speed of 2.1 GHz were used on both clusters. This means a 3-day (typical operational forecast duration) simulation will take about 0.9 hours using 1440 cores or 2.4 hours using 480 cores.

**Table 1: Baseline and sensitivity runs used in this paper.**

| Scenario | Description |
|---|---|
| **Baseline** | With forcing from tides, atmosphere, rivers (from NWM) and precipitation, initialized with HYCOM GOFS 3.1 |
| **Baseline_Wave** | Baseline with added wave effects |
| **Ocean** | Baseline forced by ocean and atmosphere (i.e., tides and storm surge) only |
| **River** | Baseline forced by rivers (i.e., freshwater inputs from NWM) only |
| **Rain** | Baseline forced by precipitation (directly on top of the domain) only |

### 3.2 Coupling with NWM

River discharges are introduced into our model at its land boundary. About 6752 intersection points are identified between the NWM river segments and SCHISM's land boundary, where the freshwater is injected as volume sources (Fig. 5). Inside the model domain, streamflow, overland flow, and precipitation are directly handled by the hydrodynamic core of SCHISM. This fully coupled configuration is rare in the existing compound flooding simulations (Santiago-Collazo et al., 2019). To ensure the accuracy and robustness of SCHISM in simulating hydrological and hydraulic processes including the overland flow, we already examined the model's performance in both lab-scale and field-scale tests in a previous study (Section 2.2 and 2.3 of Zhang et al. (2020)) and applied the model in the Delaware Bay watershed including the Delaware River (extended to 40 m above the NAVD88 datum) with a hydraulic jump (Fig. 14 in Zhang et al. (2020)). The NWM segments explicitly reproduced in our grid (Fig. 3c) during the mesh generation stage help capture the bathymetry of main flood pathways (thalwegs). However, flow is *not* restricted to these 1D segments; in fact, precipitation may generate overland flow on any 2D horizontal grid elements in the watershed. Note that the river flows injected at the land boundary have indirectly incorporated the precipitation that occurred *outside* (but not *inside*) the model domain, and therefore, the addition of direct precipitation onto the model domain is appropriate and is an integral component of the compound flood processes. To accurately simulate the initial movement of the very thin layer of rainwater on the dry land, which is dominated by friction, a very small threshold of $10^{-6}$ m is used to differentiate between wet and dry states (Zhang et al., 2020). Since we have no information on the scalar concentrations for river inflows and rainfall, we applied 0 PSU for salinity and ambient water temperature (i.e., the temperature at the local receiving cell calculated without accounting for the rivers or raindrops) for the injected river water and also for the rainfall. Obviously, the latter represents a source of uncertainty for the modelled temperature results. As explained in Huang et al. (2021), heat exchange between air and water would misbehave on such a thin layer of water in the watershed, so a threshold of 0.1 mm is set for local water depth, below which the heat exchange is

turned off. As a model limitation, infiltration is neglected in this work. In the case of Hurricane Florence induced flooding, we expect the effect of infiltration to be minor. According to NOAA's weather map[6], there was continuous rainfall along the US east coast from Sep 11, 2018 to the date of Florence's landfall (Sep 14, 2018), so the infiltration capacity of the soil was already reduced. Moreover, "wet" storms like Hurricane Florence (2018) and Hurricane Harvey (2017) tend to dump large amount of rain fall at a location for days because of the slow movement of the storm, so most of the rainfall should be on saturated soil. As another model limitation, the drainage in urban settings is not included in our model. This may have led to some occasional big errors in the predicted elevation on high water marks, for example the one large error in the urban area in Fig. 13d. We do have a plan of explicitly accounting for infiltration and drainage as volume sinks based on NWM (or other hydrologic models). However, considering the additional uncertainty this would bring, for now we choose to continue improving more important aspects of the model for operational use; the focus is on the quality of model grid, which is very likely responsible for most of the existing large errors.

The assessment of NWM calculated flow against observed flow at the two largest rivers in the region is shown in Fig. 6. Similar to our findings for other storm events, the flow produced by this particular version of NWM (v2.0) is generally consistent with USGS observation but tends to show narrower and higher peaks, with roughly the same total volume of water throughout the event (Fig. 6). The observation indicates the peak streamflow occurs about 7 days after the landfall, which is the time it takes for the rainfall induced flood to reach the coastal rivers. Note that there is typically a time lag of 1−2 days between the peak flow in NWM and the gauged flow (Fig. 6). The forcing errors in the magnitude and timing of NWM's peak flow should explain part of the model errors, especially in the watershed. For example, we found that replacing the NWM streamflow with the gauged flow at USGS Station 02109500 (Waccamaw River at Freeland, NC) improves the model skill locally. However, this is not cost-effective for our goal of operationalizing this compound flood model along the US East Coast and Gulf Coast. The developers of NWM (Gochis et al., 2018) showed that NWM's model skill was improved by each version update, with 44% of the gauges having bias < 20% in the latest version (NWM v2.0). We will adopt the newest and best NWM version as soon as it is available in our ongoing study and operational forecast. And we are open to using any other hydrologic sources to drive our model.

---

[6] https://www.wpc.ncep.noaa.gov/dailywxmap/ (last accessed in Apr 2021)

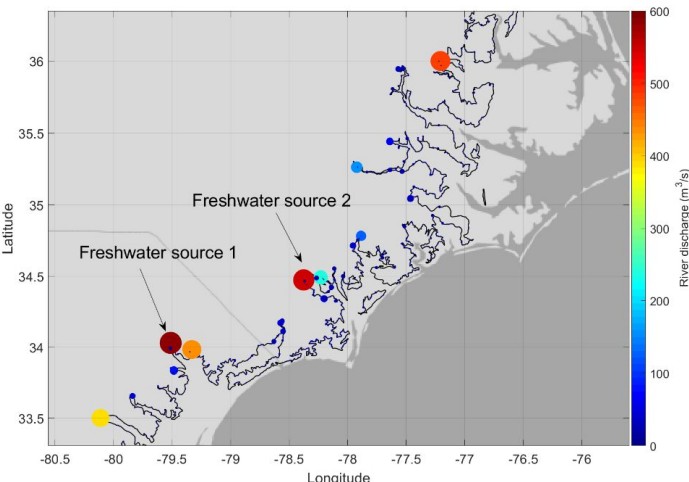

Fig. 5: Distribution of river discharge (time-averaged during the simulation period) in North Carolina and South Carolina, from National Water Model (NWM).

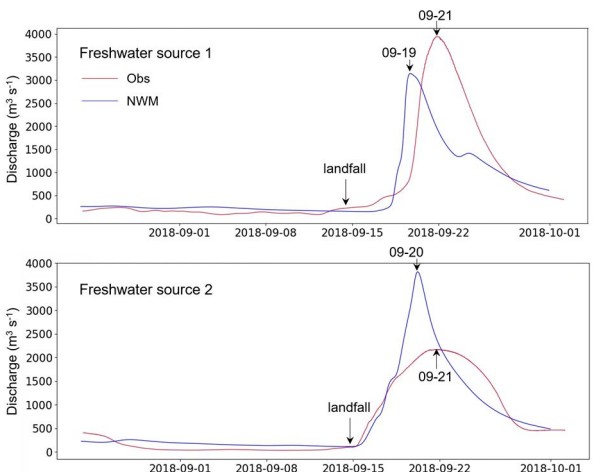

Fig. 6: Discharges at the two largest freshwater sources in the impact region (locations marked in Fig. 5). At each location, the NWM streamflow is taken at the intersection of the NWM segment and SCHISM's land boundary; the observation is based on the closest USGS station.

## 4 Model validation and sensitivity

In this section we will assess the model results for elevation, inundation and flow in the watershed and estuary. The spatial scales covered by the validation vary from O(10 km) to O(1 m). The model validation for non-storm period will not be discussed here; in short, the averaged amplitude error for the major constituent (M2 in east coast and K1 in northern Gulf of Mexico) for non-storm period is 3−4 cm (see Huang et al., 2021). We will start by looking at the wave effects nearshore.

### 4.1 Wave effects

Multiple breaches and over-toppings were reported[7] across several NC barrier islands during the event, including Surf City,
North Top Sail Beach, and New River inlet. Some of these breaches may be related to significant wave activities, for
example, the maximum wave height at buoy 41025, ~30 km offshore from Cape Hatteras, reached ~10 m. Interestingly, the
maximum wave heights become relatively modest nearer to the landfall: 4–6 m at Buoys 41159 and 41013 (see Fig. 3b for
their locations). Therefore, to investigate this possibility, we have also conducted a simulation with the wave model inside
SCHISM activated ("Baseline_Wave" in Table 1). The details of the wave model (Wind Wave Model) have been described
in Roland et al. (2012) and the coupled model has been applied to other systems (Guérin et al., 2018; Khan et al., 2020). The
wave model was initialized and forced at the ocean boundary by a global Wave Watch III simulation[8], and used a spectral
resolution of 36 directional bins and 24 frequency bins to cover a frequency range of 0.04 to 1 Hz. The coupling time step
between the two models (i.e., the interval at which the information of surface elevation, velocity and wave radiation stress
was exchanged) was 600 seconds.

Our results indicate that the barrier islands near Surf City, North Top Sail Beach and New River inlet were indeed over-
topped with 1–2 m of water (Fig. 7b; the locations of the islands can be seen in Fig. 8b). On the other hand, large portion of
the island north of Cape Hatteras (Outer Banks) was spared (Fig. 7c), likely due to its N-S shoreline orientation, even with
the 10 m wave approximately 30 km offshore from there. More quantitative validation for the breaching processes is out of
scope here because (1) we do not have accurate and update-to-date bathymetry just before the event, and (2) more
importantly, a sediment transport study is required to simulate the bathymetric changes.

The wave effects on the surface elevation are further quantified in Fig. 8, which suggests that the effects are most
pronounced (with 30 cm or larger differences) inside the estuaries and Albemarle-Pamlico Sound (APS) due to large wave
breaking nearby. In the intermediate and deep water, however, the wave effects are on the order of a few centimetres and
negligible (Fig. 8a). The Baseline results (without wave effects) also showed over-topping of the barrier islands similar to the
Baseline_Wave (not shown).

---

[7]        https://www.wusa9.com/article/weather/before-and-after-hurricane-florence-changes-north-carolina-coastline/65-
595918389 (last accessed in Apr 2021)
[8] ftp://ftp.ifremer.fr/ifremer/ww3/HINDCAST(last accessed in Jan 2021)

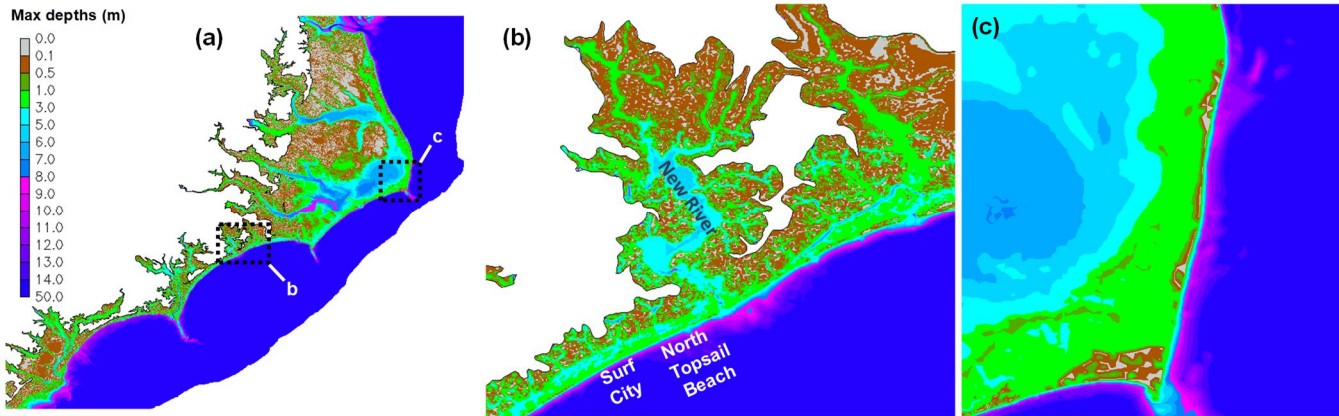

**Fig. 7: The maximum water depths from Baseline_Wave. The spatial extents of the local regions of (b) and (c) are marked in (a).**

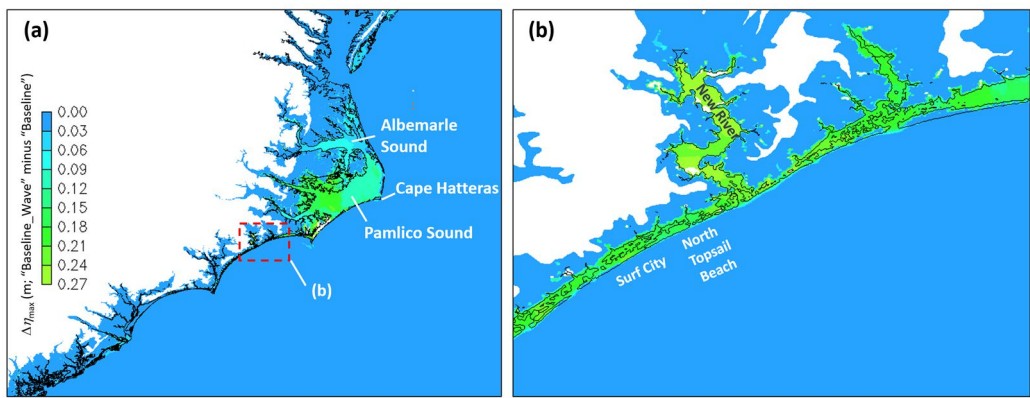

**Fig. 8: Differences of the maximum elevations between Baseline_Wave and Baseline. (b) is zoom-in from (a). The thin black lines are the 0 m isobath.**

In summary, the wave effects are significant nearshore and have contributed to the observed breaching and over-toppings of barrier islands. However, because the current study does not focus on the breaching processes and because of the

285 significant computational overhead introduced by adding the wave model (>50%), we will proceed in the following by using the run without waves as the "baseline".

### 4.2 Bays and estuaries

We assess the calculated water levels at six NOAA tide gauges near the impact area (see locations in Fig. 3b). Three gauges are facing the open ocean (Charleston, Springmaid, and Wrightsville), and the other three gauges are either sheltered

inside a bay (Beaufort) or behind barrier islands (Hatteras and Oregon Inlet). The responses to the hurricane are different at those gauges, as seen from the total elevation (Fig. 9a) and the sub-tidal signals (Fig. 9b). The latter applies a low-pass Butterworth filter (Butterworth, 1930) only preserving longer-period (longer than 2 days) components. The observation shows see set-downs of ~0.3–0.5 m at Charleston, Springmaid, Hatteras and Oregon Inlet followed by surges of ~0.2–0.5 m;

and surges of ~0.5 to 1 m at Wrightsville and Beaufort (Fig. 9). The different responses at these gauges are due to the wind curl of Florence that led to different dominant wind directions between southern and northern stations, and are also due to specific geographic settings of each gauge. Most intriguing are the prominent set-downs observed at Hatteras and Oregon Inlet, which are explained by a combination of wind direction and blocking effects of barrier islands. Fig. 10 demonstrates that around the time of landfall of the hurricane, the predominantly westward wind felt in the Pamlico Sound has pushed water away from the barrier island. Meanwhile, the surge that propagated from the ocean side is effectively blocked by the barrier island chain, thus creating a ~70 cm elevation difference between the waters immediately outside and inside the islands (Fig. 10). The mechanism causing the water level set-downs at the two South Carolina stations (Charleston and Springmaid) is similar to that causing the set-downs behind the barrier islands in North Carolina. The two South Carolina stations are located to the south of the landfall site and the wind direction is from the land to the ocean, pushing water away from shore. The model captured the different regional responses; overall, the averaged MAE (mean absolute error) for elevation is 11 cm. The averaged MAE for the subtidal comparison is 8.6 cm and averaged correlation coefficient is 0.92. The peak errors at different stations occur around the storm surge, with a maximum over-prediction of 0.64 m for the peak surge at Spring maid Pier, SC. The overpredicted peak surge can lead to overpredictions in elevation on coastal high water marks (HWMs). In addition, there is a maximum under-prediction of 0.66 m for the set down at Hatteras, NC, mainly due to the mismatch in the set down timing. The uncertainties in wind forcing may be the main cause of the error, which is predominantly from subtidal signals. The grid quality near the barrier islands may also contribute to the error. Adding wave effects slightly increases the surge and rebounding waves at the last three gauges, resulting in slightly better model skills there.

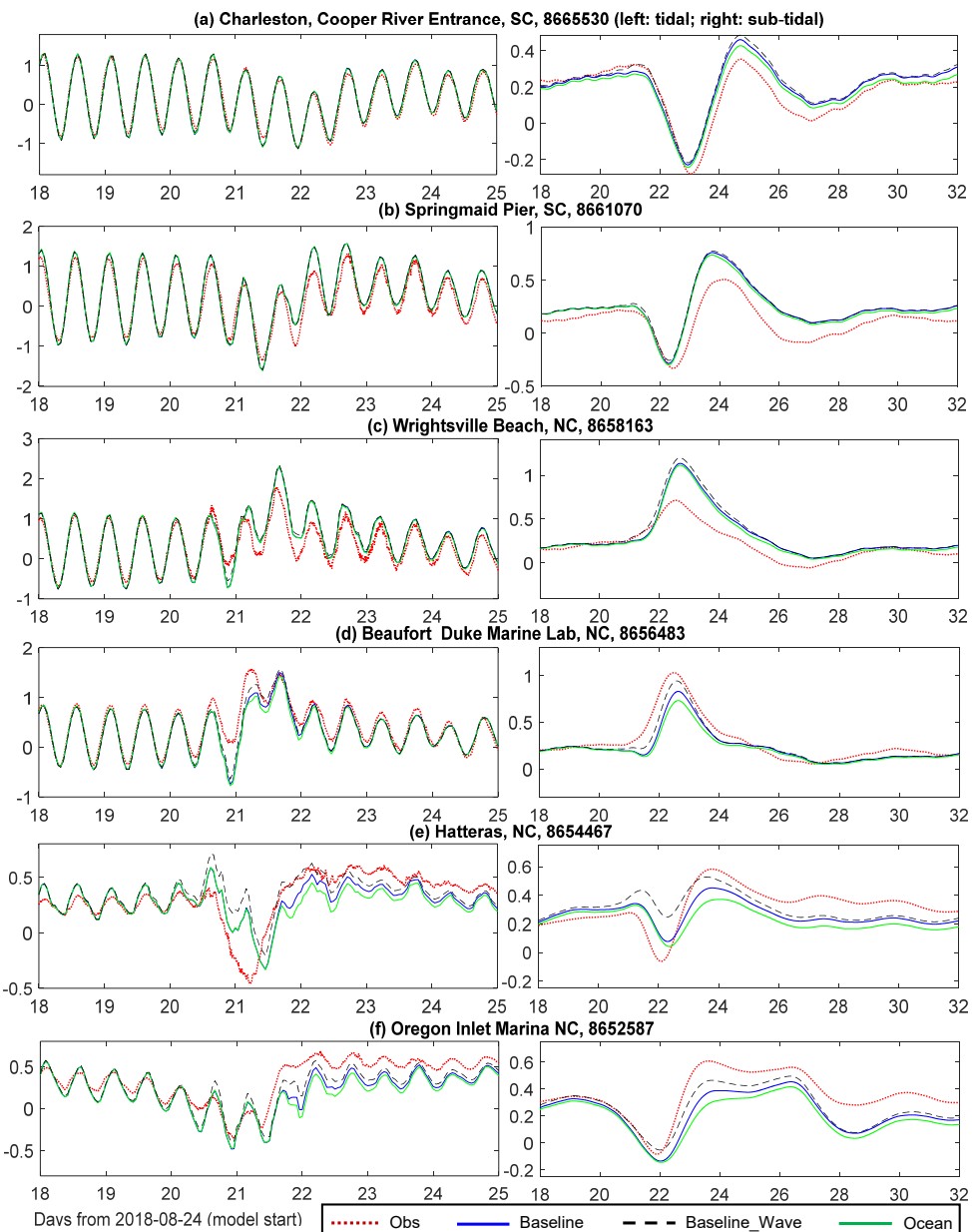

**Fig. 9: Comparison of elevation at six NOAA gauges: (left) total elevation; (right) subtidal elevation. Also included are results from two sensitivity runs ("Baseline_Wave" and "Ocean"; see descriptions in Table 1). Note the plots have different y-axis ranges.**

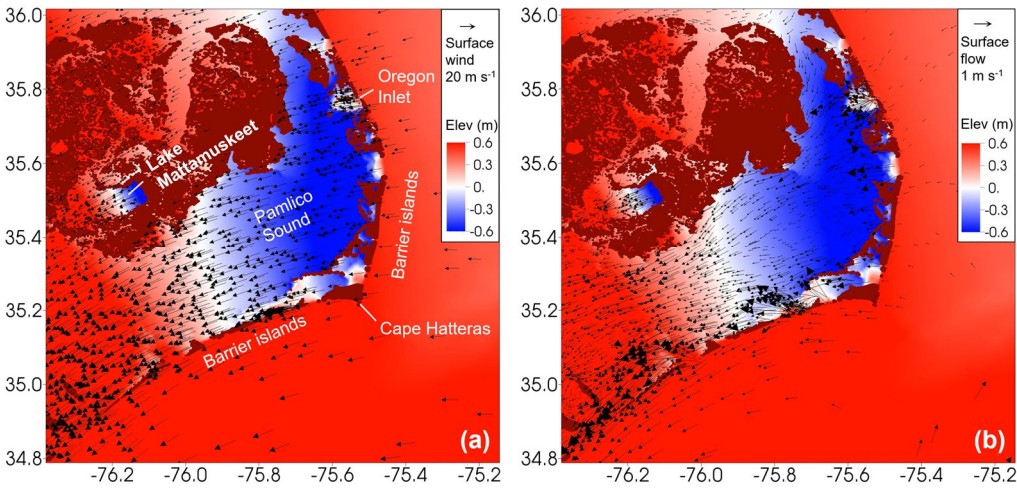

**Fig. 10: Snapshots of water surface elevation at the time of Florence's landfall (2018-09-14) near the barrier islands around Pamlico Sound, overlaid by (a) wind speed at water surface and (b) surface flow. Note that the general pattern of wind-induced set-up/set-down is also present on a smaller-scale (Lake Mattamuskeet).**

## 4.3 Watershed

High Water Marks (HWMs) were collected by USGS experts more than two weeks after Hurricane Florence's landfall. They are derived from small seeds or floating debris carried by floodwaters that adhere to smooth surfaces or lodge in tree bark to form a distinct line, and also by stain lines on buildings, fences, and other structures. Therefore, HWMs are time sensitive and usually have vertical uncertainties of ± 0.3 feet, or equivalently ± 0.09 m (Koenig, et al., 2016; Austin, et al., 2018).

The simulated elevation on 276 HWMs in the NC and SC watersheds are compared with field estimates (Fig. 11). The model is able to capture the transition from estuarine to riverine regimes; note that the averaged bottom elevation for all observation points is 3.8 m (NAVD88), and about 70% of the points are located above 2 m (NAVD88), beyond the reach of storm surges. Overall, the averaged MAE for all HWMs is 0.73 m, with a correlation coefficient of 0.92 and a positive mean bias of 0.09 m. There is a slight positive bias on near-shore HWMs (Fig. 11), corresponding to the over-prediction of peak 335    elevation at coastal stations (Fig. 9a-c). These skill scores are similar to what we obtained for Hurricane Harvey (Huang et al., 2021).

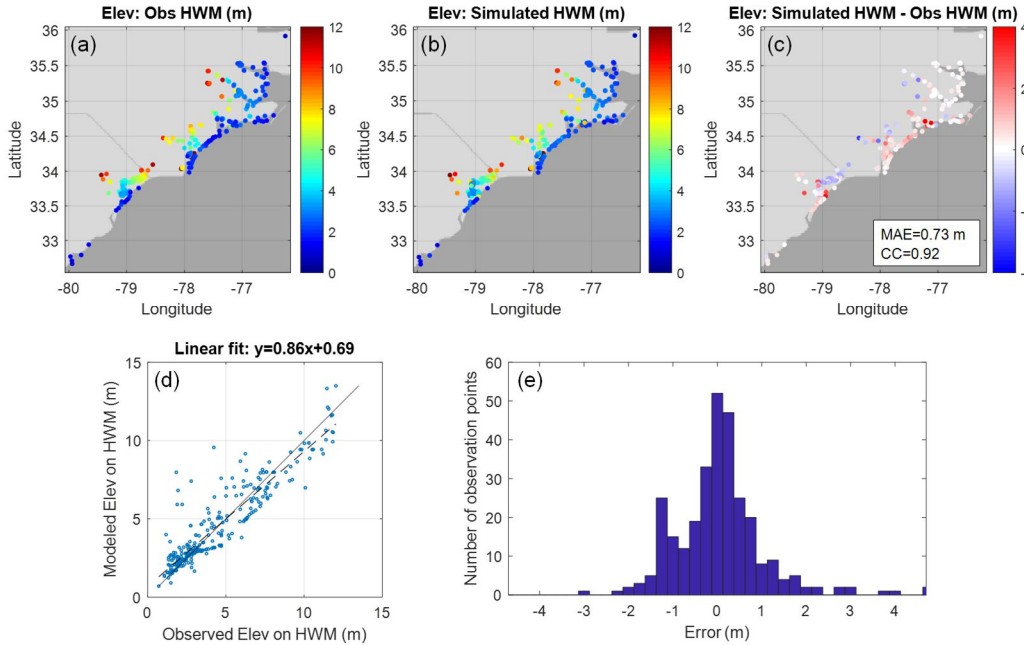

**Fig. 11: Comparison of water surface elevation on HWMs. (a) Field estimates from USGS; (b) model prediction; (c) model minus observation; (d) regression between model prediction and observation; (e) histogram showing the error distribution.**

Fig. 12 shows the comparison for both elevation (gauge height) and discharge in a large river in the study region. The gauge is in the interior of our grid, near "Freshwater Source 2" in Fig. 6. Because the observation's vertical datum is NAVD29 and the model's datum is NAVD88, we have adjusted the mean model elevations to match the observed mean in the elevation comparison (Fig. 12b); in other words, only the elevation variability is compared. Our model over-predicted the flow and under-predicted the flood-induced surges. Using more accurate fresh water source at the land boundary, improving

the channel representation in the model grid, and locally adjusting the bottom friction should help improve the skill.

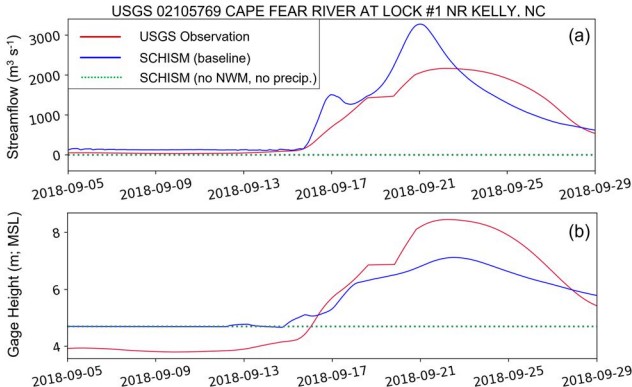

**Fig. 12: Model-data comparison near the landfall location: (a) streamflow; (b) gauge height. The sensitivity run (without freshwater inputs from NWM or precipitation) is also shown, in which the channels are dry even during the hurricane, indicating they are beyond the storm surge influence. The station location is marked in Fig. 3b.**

Our tests show that the simulated elevation on the High Water Marks (HWMs) in the watershed is sensitive to: grid resolution, precipitation, river inputs through the land boundary, and bottom friction. Grid resolution and quality are the most important factors. Misrepresenting flood pathways can easily lead to errors of a few meters near some very localized features such as ditches and highways. Fig. 13 illustrates such an example around the Burnt Mill Creek in the city of Wilmington, NC. Large HWM errors were found in the preliminary setup, because the computational grid did not resolve the small creeks that served as the main conduit in draining out the storm water after the flood. This resulted in stacking of water locally and thus large over-prediction of HWMs there. The channel of the creek is about 6–10 m wide, and once resolved using 2 rows of quadrangles as done in the baseline (Fig. 13c), the model skill was greatly improved (Fig. 13d). The only remaining large error point in the "baseline" occurs in an urban area away from the river (Fig. 13d), likely due to the building or drainage effects that have not been incorporated in the model. The defects in grid quality can lead to large errors that are not likely to be rectified by tuning other parameters. To fix the remaining few large errors away from the landfall site, grid quality should be examined first. The continuous improvement on this model grid is part of an ongoing effort of operationalizing the model along the US East Coast and Gulf Coast, and we will report this in future studies. Resolving small-scale flow routing features in a national scale requires automated tools such as Pysheds[9] that can detect and delineate the channels automatically. Initial tests showed very promising results from this package. We remark that it is feasible to resolve these features efficiently without significantly increasing the grid size due to SCHISM's flexibility and robustness in handling poor-quality meshes. Afterwards, the inclusion of urban drainage should reduce the occasional large errors there. Other factors such as uncertainties in DEM, precipitation, and the river flow through land boundary also play minor roles.

---

[9] https://github.com/mdbartos/pysheds (last accessed in Jan 2021)

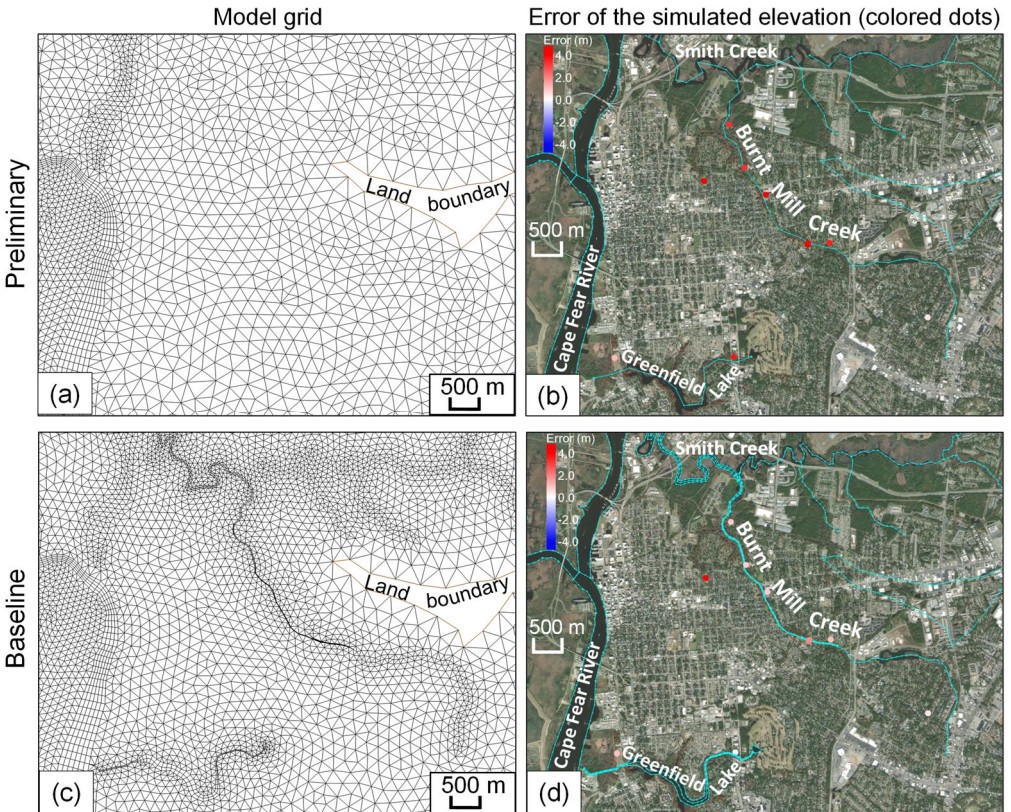

**Fig. 13: Importance of resolving small-scale features on the order of a few meters in the watershed, illustrated by a comparison between a preliminary setup (a, b) and the baseline setup (c, d). To better resolve the Burnt Mill Creek, NC, more SMS feature arcs (cyan lines in (d)) are used in the baseline setup than in the preliminary setup (cyan lines in (b)), significantly reducing the HWM errors. See Fig. 2 for the location of this locally zoomed-in region. The base maps in (b) and (d) are provided by ESRI.**

## 5 Compound effects

A carefully validated 3D model such as the one presented here can effectively separate out compounding factors from different sources: coastal surge, river flooding and precipitation. In this section we apply this approach to examine the contributing factors to the total flooding during Florence. The design of the numerical experiments is such that we selectively turn on/off forcing from ocean, river, and precipitation to examine their individual effects in isolation (Table 1). As an overview, the conditions of maximum inundation extent from all scenarios are listed in Table 2. To facilitate the comparison of inundated area, a practical value (1 foot) on the same order of the mean inundation depth is used as a threshold. For the two indices (percent inundated area and maximum inundation depth) shown in Table 2, the baseline values are significantly larger than those from a single sensitivity test. This confirms the existence of compound regions in the two states (North Carolina and South Carolina) during the event. More details of each forcing's effect and the compound effects are discussed below.

**Table 2: Overview of the maximum inundation extent in South Carolina and North Carolina watersheds (above the NAVD88 datum) during the simulation period.**

| Scenario | Percentage of inundated area with water depth > 0.305 m (1 foot) | Spatially averaged maximum inundation depth (m) |
|---|---|---|
| Baseline | 46.7% | 0.61 |
| Ocean | 12.7% | 0.12 |
| River | 17.4% | 0.31 |
| Rain | 34.4% | 0.36 |

Turning off both rivers and precipitation (i.e., ocean only) is expected to have major impact on flooding in the watershed. This is confirmed in Fig. 12 in the previous section. Not surprisingly, without rivers and precipitation, watershed is mostly dry as the storm surge cannot propagate over steep terrains. As a result, the predicted HWMs are biased too low (Fig. 14) as the steep topography quickly damped out any surges brought in by the ocean. This leads to systematically underpredictions in the watershed and a 64% increase in MAE compared to the baseline (Fig. 11). On the other hand, Zhang et al. (2020) demonstrated that the storm surges can propagate much further into watershed if watershed rivers are included. There is no apparent deterioration of model skill on the near-shore HWMs, because those locations are predominately affected by oceanic processes.

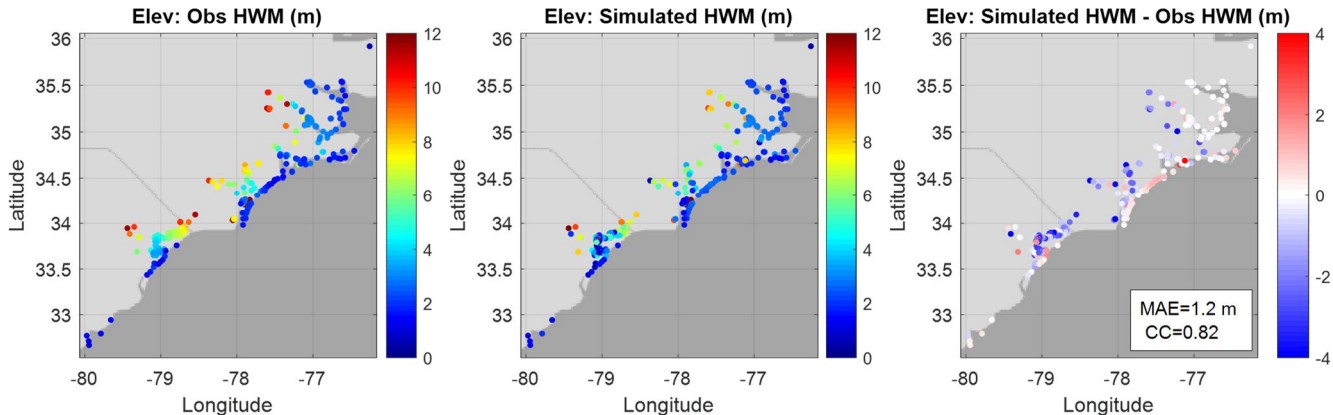

**Fig. 14: Simulated water surface elevation on HWMs from the sensitivity run "Ocean", i.e., without the freshwater inputs from NWM or precipitation. Note the underpredictions in the watershed and worse model skill compared with "baseline" (Fig. 11).**

Less obvious are the effects of rivers and precipitation on the observed surges in the coastal bays. Fig. 9 indicates that the effects are negligible at the three coastal stations away from barrier islands (Fig. 9a–c), as the large amount of freshwater from the watershed directly drains into the coastal ocean (which has much larger volume of water). On the other hand, the

impounding effects are clearly seen at the two stations behind the barrier islands (Fig. 9ef; roughly starting from Day 22), where the discharged water is trapped for almost a week. As we showed in Section 4.2, the barrier islands are effective in
creating separation and thus large elevation gradient between water immediately outside and inside (see Fig. 10).

To assess the contributions from each of the 3 forcing factors to the total sum, we follow Huang et al. (2021) and use the concept of "disturbance". We recognize that for compound flooding processes involving both ocean and watershed, neither the water surface elevation nor the water depth is a satisfactory metric, because the nominally large water elevations on the high ground of watershed are dominated by the high bottom elevation there, and the large water depths in the bays and ocean
are dominated by the local bathymetry. Therefore, we adopt the concept of "disturbance", defined as:

$$D = \begin{cases} \eta, & \text{if } h \geq 0, \\ \eta + h, & \text{if } h < 0, \end{cases}$$

where $\eta$ is the water surface elevation and $h$ is the bathymetry (positive downward based on the same datum as $\eta$; e.g., $h > 0$ for ocean and $h < 0$ for high grounds in watershed), so $(\eta + h)$ is water depth. Basically, $D$ represents the departure from "initial condition" (either the initial water surface or bottom, whichever is higher). Note that $D$ is continuous across $h = 0$.
On the initially "dry" ground in watershed, $D$ represents the local water depth; whereas at initially "wet" locations, $D$ is simply the surface elevation. $D$ is also a smoother metric to measure the compound effects as one transition from oceanic into watershed regimes.

The comparison between the maximum disturbances from the three experiments using only one of the three forcing factors and the total sum helps elucidate the contributions from each. The results are presented in Fig. 15, in terms of proportion of
total maximum disturbance as explained by each forcing factor at a given location (therefore, the sum of all proportions equals unity). Our results clearly indicate that: (1) the ocean (and atmospheric) forcing dominates in the open and part of southern Pamlico Sound in the APS system; (2) river forcing is most dominant in the river network of watershed; (3) precipitation effects are dominant in other parts of the watershed away from the river network. However, the presence of the barrier islands significantly complicates the interaction among different forcings (e.g., the significant contribution from
precipitation in Pamlico Sound as shown in Fig. 15c). On the other hand, the inclusion of the wave effects is not expected to alter the findings here because their contribution to the total elevation is relatively minor as compared to the atmospheric effects (see Fig. 9).

The competition among different forcing factors in different regions can be succinctly summarized in a "dominance" map as shown in Fig. 16: a factor is deemed dominant if it explains at least 80% of the total disturbance; if, however, none of the
three factors contribute to 80% or more at a particular place, the nonlinear compound effects are expected to be significant there. The ocean response is overwhelmingly dominated by the oceanic and atmospheric forcing, the response in the watershed rivers by the river flow, and the response in large portion of highly-elevated watersheds by the precipitation, as seen in Fig. 16. Most of the response in the southern Pamlico Sound is of oceanic origin, because of the wider openings to the south (e.g., Ocracoke inlet). On the other hand, there are only a few narrow inlets to the east (e.g., Oregon Inlet), thus

effectively blocking off the oceanic influence there (Fig. 16b; also see Fig. 10). It is the "grey areas" (Fig. 16) of compound flooding zones that are most intriguing. These include most of the APS and coastal rivers (Fig. 16b), where the weakened oceanic influence competes with river flow and large rainfall there (Fig. 15). In the estuaries with large river discharges and limited openings to the coastal ocean (Fig. 16c & e), the compound flooding zone is a result of the competition among all three factors. On the other hand, the estuaries to the south of the landfall site have smaller river discharges and less

precipitation, moreover, they are not protected by barrier islands. As a result, oceanic effects can penetrate very deep into these coastal watersheds (Fig.16d). Note that the ocean dominance near Wrightsville Beach, NC (NOAA Station 8658163) and Springmaid Pier, SC (NOAA Station 8661070) may be exaggerated, considering the overestimated peak elevation there (Fig. 9). The compound map in Fig. 16 clearly demonstrates the urgent need for a holistic management approach in the planning of mitigation effort for the flood hazard during and after hurricane events.

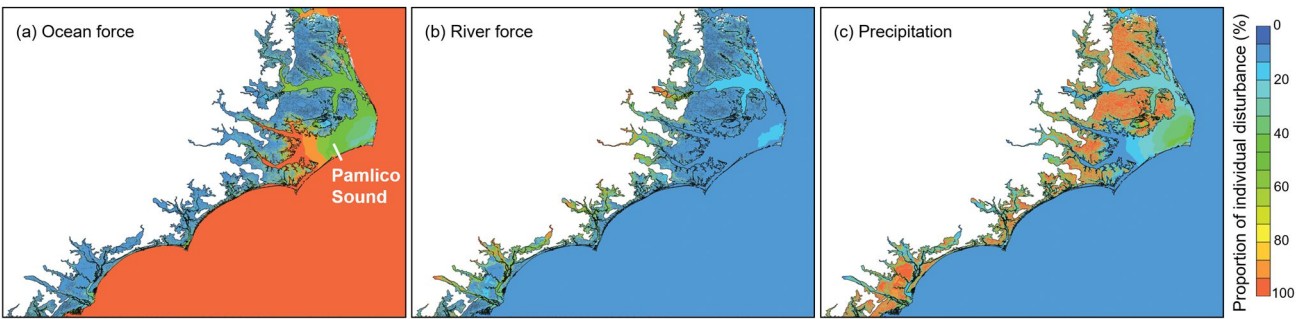


**Fig. 15: Regional map showing the spatially varying importance of each forcing factor: (a) ocean force ("Ocean" in Table 1); (b) river force ("River" in Table 1); and (c) precipitation ("Rain" in Table 1). The value is the proportion of a factor's individual "disturbance" (see definition in Section 5) to the sum of the disturbance from all factors. The colors from blue to red represent increasing importance of a factor at a specific location.**

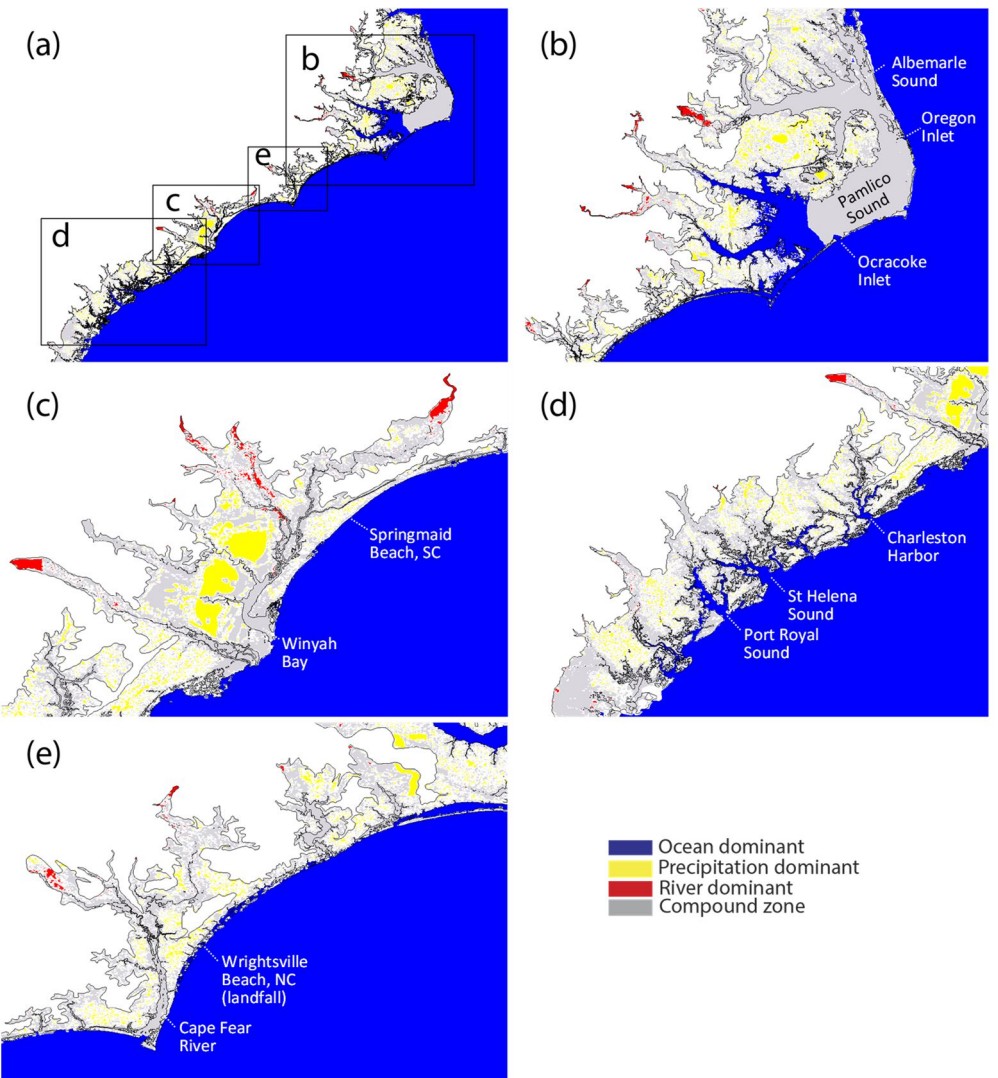

**Fig. 16: Dominance map showing the spatial dominance of different flood drivers during Hurricane Florence. (b−e) are zoom-ins from (a) in different regions.**

## 6 Conclusions

We have successfully applied a 3D cross-scale model to examine the compound flooding processes that occurred during Hurricane Florence 2018. The model is fully coupled in the sense that the hydrologic and hydrodynamic processes are solved by the same set of governing equations. Limitations of the model include the neglection of infiltration and urban drainage, which will be implemented soon. The model was validated with observation data collected in the watershed and coastal ocean. The mean absolute errors for major variables are: 11 cm for coastal elevation, 72 cm for High Water Marks (HWMs). Locally very high resolution was used in some watershed areas to resolve small features that were critical for a good model

skill for the HWMs. The wave effects were found to be significant near barrier islands and had contributed to over-toppings and breaches there. The validated model was then used to reveal significant nonlinear compound effects in most parts of coastal watersheds and behind the barrier islands. The barrier islands were shown to be particularly effective in separating the processes in the water bodies on the land side and on the ocean side.

The results of the current study, especially the regional compound zone map, filled in a critical knowledge gap in our understanding of compound flooding events. In fact, operational forecasts based on the current model are being set up at NOAA to help coastal resource and emergency managers with disaster planning and mitigation effort. The model can also be used to facilitate new scientific discovery of novel coastal processes; for example, preliminary results for the fate of pollutants discharged from the watershed suggest that the large watershed outflow resulted from heavy precipitation played

an essential role in exporting pollutants far into the ocean through the large and long-lasting freshwater plumes that occurred after the event.

**Data availability**

The model source code is freely available at https://github.com/schism-dev.

**Author contributions**

All authors conceived the idea of the study under the NOAA's Water Initiative project; FY, WH and YJZ developed the methodology with the support of other co-authors; FY and WH produced the results with the support of YJZ, HY; FY analyzed the results with the support of other co-authors. All authors contributed to writing of the paper.

**Competing interests.**

The authors declare that they have no conflict of interest.

**Acknowledgements**

This work is funded by NOAA's Water Initiative (Grant Number NA16NWS4620043). The authors thank Dr. Linus Magnusson (ECMWF) for providing the high-resolution ERA forcing. Simulations used in this paper were conducted using the following computational facilities: (1) William & Mary Research Computing for providing computational resources

and/or technical support (URL: https://www.wm.edu/it/rc) (2) the Extreme Science and Engineering Discovery Environment (XSEDE; Grant TG-OCE130032), which is supported by National Science Foundation grant number OCI-1053575; (3) the NASA High-End Computing (HEC) Program through the NASA Advanced Supercomputing (NAS) Division at Ames Research Center.

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
