# Peer review of "A cross-scale study for compound flooding processes during"

_Natural Hazards and Earth System Sciences, 2020_

## Referee Comment (RC1) · Anonymous Referee #1 · 15 Dec 2020

Thank you for this opportunity to review this manuscript. This is a very well writing manuscript. The authors present an interesting compound flood case study during Hurricane Florence (2018). I enjoyed reading the manuscript. However, I have a few questions/comments as listed below.

1. The model is not calibrated. The authors selected a few constant values as the friction values at various locations in the models based on previous studies. Although the authors tried to justify this by saying "in favour of simplicity", I am not convinced that a model without calibration will be of great value. 2. Is the NWM model calibrated? If yes, please give details. 3. With such detailed 3D modelling of such a large area, what the efficiency of the model is like? E.g. what is the computational time for a flood event lasting for a specific period of time (e.g.3 days)? What is the specs of the computing

[Figure]

facility used? 4. Line 252: The authors reported the average MAE. But often it is the peak error that is important. What is the peak error, when and where did it occur? What is the potential impact of this peak error? 5. Figure 11: The figure shows modelling errors of up to $\pm4$ meters at various locations. Do the authors have an explanation on the large errors at these locations (apart from just saying calibration can improve model performance)? - The authors did a good job explaining model performance in relation to grid resolution. A similar explanation here will be good. 6. What is the return period of flooding at various locations during this event? A comparison on the return period for floods caused with or without compounding effect will give readers a clearer picture of the impact of the compound effect. 7. Figure 14: Please see comments on Figure 11.

Minor comments: 1. Figure 15. I understand this figure is used to show the impact from different compound flood driver. Ii is difficult to interpret the results. The caption can include some accompany text on how the figure can be interpreted (what the proportion values imply).

---

## Author Comment (AC1) · 17 Jan 2021

Dear Reviewer,

Thank you for taking time to help us improve the manuscript. We agree on many points you raised, and we will address them in a later reply and in the revised manuscript. Before that, we would like to briefly comment on a few points.

<Reviewer Comment #1>

We agree with the reviewer that "The tuning of friction values is intentionally kept to minimum in favour of simplicity" is not a good statement. We will add more details on calibration in the revised version.

[Figure]

Calibration was indeed an important step during our model setup. The friction of the baseline model was tuned in wet area (river, estuary, ocean; lower than 1 m above MSL) and on higher grounds (higher than 3 m above MSL) separately. In the wet area, drag coefficients were tested within a range of 0.001-0.01. Commonly accepted default value of 0.0025 gave good error statistics near the landfall site; but as you can see from the attached figure, there is not a lot of sensitivity in the range of 0.001-0.005.

In the watershed, drag coefficients were tested within a range of 0.01-0.5. The optimal value was chosen based on the High Water Marks (HWMs) comparisons at over two hundred locations, collected by USGS. A small friction value within this range tended to under-predict the elevation at HWMs, and a large value led to over-prediction. Values with the range of 0.02-0.05 gave good error statistics. We chose 0.025 because it gave slightly better results in the Cape Fear River watershed near the landfall. Note that this is the parameterization adopted for the region influenced by Florence. Spatially varying parameterization of bottom friction for different systems is an on-going effort as we study more recent hurricanes and operationalize the model along the East Coast and Gulf Coast. However, as we presented in Ye et al. (2020), Zhang et al. (2020) and Huang et al. (2021), the choices described above seem to work fine in general for other systems as well.

Note that we have already published extensive model validation and sensitivity test results with respect to other parameterizations (e.g., 2D vs 3D etc.) in the above-mentioned citations. We feel that we should focus on new findings in this paper instead of presenting similar findings once again.

<Reviewer Comment #3>

For the baseline run, the real time ratio to simulation time is 80 with 1440 Intel Skylake cores on TACC's Stampede2 and 30 with 480 Intel Skylake cores on W&M's Sciclone. So, a 3-day simulation will take 0.9 hours using 1440 cores or 2.4 hours using 480 cores. Although the model covers a large domain, most of the elements are 2D, making

it efficient enough for operational forecast. We will add this information into the revised manuscript.

<Reviewer Comment #5>

This is a very good point, and we will add more explanations into the revised manuscript. In short, the few large errors are likely the result of similarly under-resolved bathymetric/topographic features, with minor contributions from other factors such as DEM uncertainties.

Our tests show that the simulated elevation on the High Water Marks (HWMs) in the watershed is sensitive to: grid resolution, precipitation, river inputs through the land boundary, and bottom friction. Grid resolution and quality is the most important factor, as explained in the last paragraph of Section 4.3. Misrepresentation of flood routing can easily lead to errors of a few meters near some very localized features like ditches, highways etc. How to resolve all important small features efficiently is an on-going research area.

The defects in grid quality can lead to large errors that are not likely to be rectified by tuning other parameters. To fix the remaining few large errors away from the landfall site, grid quality should be examined first. The continuous improvement on this model grid is part of an ongoing effort of operationalizing the model along the US East Coast and Gulf Coast, and we will report this in future studies. Other factors such as uncertainties in DEM, precipitation and the river flow through land boundary also play minor roles.
* * *
[Figure]

**Fig. 1.**

---

## Referee Comment (RC2) · Anonymous Referee #2 · 22 Jan 2021

Congratulations to the authors on putting together an interesting and well-written manuscript. The study seeks to understand the role of various coastal flood mechanisms occurring during Hurricane Florence (2018), such as tides, storm surge, waves, river flows and rainfall. The authors utilize a 3D hydrodynamic model (with river input from the National Water Model) to simulate the combined flooding and investigate the individual contributions of each major flood driver during Florence. In general, this is a well-written and worthwhile contribution to the field of compound flooding literature. I have some comments that I believe can improve the manuscript, and I recommend acceptance of the manuscript after the authors have considered the main points outlined below:

- The authors suggest that a 3D baroclinic model is necessary to accurately capture the

water level response. However, I suspect that utilizing such a complex model (plus the very large mesh size) would require a very large computational expense. The authors do not present results from using a more simplified approach (2D depth-averaged for example), so we don't know whether the additional complexity is actually needed to capture the water level response. The manuscript would benefit from some discussion about the trade offs between model complexity and computational burden, as well as some details about how long each simulation takes to run.

- In my experience, the NCEI CUDEM does not capture the bathymetry of coastal streams well, and tends to significantly underestimate the channel depth. Did the authors make any corrections/modifications to the raw DEM to better represent coastal streams?

- There is no mention of infiltration in this manuscript, which is an important factor controlling the pluvial flood dynamics. Did the authors completely neglect infiltration in this work? And if so, there should be some discussion/justification about why infiltration was not accounted for. Since much of the underlying soil in the NC region is sand, I expect infiltration losses would be non-negligible and neglecting infiltration could cause over-estimation of the rainfall-induced flooding.

- In addition to these comments, I have pointed out a couple issues/suggestions for the figures in the manuscript (see specific comments).

Specific Comments:

Line 106: What is meant by "bona fide" in this context? Please clarify.

Figure 3: I am a bit confused by figures 3b and 3c. The box showing the location of figure 3c indicates that 3c is north of Albermale Sound. However, when I look at Figure 3c it appears to be depicting the Cape Fear River Estuary, which is at the southern tip of NC. Please confirm/update the actual location of Fig 3c.

Table 1: The difference between the No_NWM_precip and Ocean scenario is not clear

[Figure]

to me. It seems they both do not include river or precip forcing?

Fig 9: It is difficult to distinguish the differences between the model runs, especially for figures 9a-d. I suggest the authors shorten the depicted time window to only show 1-2 days before landfall through 4-5 days after landfall for a-d. This will allow a reader to see more clearly the differences between the scenarios and the comparison to the observed water levels.

[Figure]

---

## Referee Comment (RC3) · Anonymous Referee #3 · 8 Feb 2021

Thanks for the opportunity to review this manuscript. The article presents a coupled hydrologic-hydrodynamic modeling framework to simulate compound coastal flooding considering the impacts of various flood drivers (coastal, fluvial and pluvial). The SCHISM is used for baroclinic simulation (with vertical grids) of coastal processes. To account for inland processes the authors have used National Water Model (NWM) outputs at the land boundary (set at 1m above sea level) and used ERA re-analysis products for rain-on-grid simulations to hindcast flooding dynamics due to Hurricane Florence. They further run their model under baseline and sensitivity runs to quantify the contribution of different sources of flooding in various compound flood hazard situations. The idea of integrated modeling that couples detailed hydrologic and hydrodynamic modeling schemes to simulate the nonlinear behavior of interactions between

various flood sources at coastal regions is interesting and is currently going on in various research groups around the globe. To me the novel aspect of this work is having very much detailed representation of hydrology (via NWM based on WRF), river morphology and coastal ocean dynamics (via a 3D baroclinic model). The manuscript is well written and can be of interest of community, thus I suggest publication in NHESS, however after a major revision. The major issue to me is that, despite the efforts of authors to calibrate/validate their model, on the hydrologic fluvial side the model does not seem well calibrated, and the significant error in NWM prediction (both timing and magnitude of peak) can pose a significant error to the overall estimates. Also, not clear to me how the pluvial flooding processes (from direct rainfall) is treated here. More detailed comments below:

Major comments: - Hydrologic processes are not well represented here. River flow hydrograph estimates from NWM are significantly different than the observed flow at USGS gauges (see Figure 6). About 2-3 days of lag in peak flow estimation and up to 100% error in peak flow magnitude can pose a significant error in overall inundation and coastal water level dynamics if propagated through the system. Timing (and magnitude) of peak runoff plays a significant role in extreme water level dynamics in freshwater-influenced coastal systems and getting these hydrologic characteristics right plays a major role in accurate estimation of extent/intensity of compound flooding. This might partially explain the wide range of difference (up to 4 m) between simulated and estimated high water marks (Figure 11), with a considerable number of points having estimation error greater than 1m (Figure 11e).

- Could not find any information on how the pluvial processes are handled in the model. Rain-on grid modeling (L174) is a good contribution to the field which I acknowledge, but how the underlying processes and variables that contribute to the runoff generation are accounted for is vague. How infiltration capacity is accounted for? How drainage infrastructure contribution is accounted for (in urban settings)? How the generated runoff is routed between pixels? This is even more important when the results suggest

that flooding in a significant portion of coastal land in dominated by "precipitation only" (Figure 15c).

Minor comments: Figure 1: red text (in the left panels) is not readable in print. I'd also add some space between rows in the legend. Figure 3: Black text is really hard to read on the dark blue background. Figure 4 is confusing. Hurricane track is numbered right-to-left (Panel a) and vertical grid points are numbered Left-to-Right (Panels b and c) L110: what "LSC" stands for? L120: "10 m above sea level", you mean mean sea level? if yes, is it regional or global. Please be more specific about this. L133: what "CFL" stands for? L143: Explain what is HYCOM first time use it in the manuscript. L174: Simply said rainfall is routed. Please, elaborate more on this, how exactly routed? How infiltration capacity incorporated? L189: "4-6 days" in observed or modeled events? The modeled and simulated peaks seem to be few days apart themselves. Figure 9: Better to compare detided signal here. These gauges are extremely tidal influenced and estimation skills of the model for stochastic processes is diluted by highly predictable astronomic tides. L275 & L284: describing errors "quite satisfactory" and "reasonable agreement", when mean absolute error in estimated high water levels is 0.73 m and with nearly 20% of points having error >1m, is a subjective description. I'd avoid such statements and simply let the readers decide whether if such estimation errors are reasonable according to their required accuracy and project objectives.

---

## Author Comment (AC2) · 15 Mar 2021

Dear Reviewer,

Thank you for your time and effort in helping us improve our manuscript. Here, we respond to all your comments, with your original comments repeated for your reference. Some of the responses have texts/figure copied from "AC1: Short comments in reply to Review #1", which are labeled as "repeating AC1". Following the journal's publishing guideline, the revised manuscript is not uploaded at this time, so all line numbers and figure numbers are based on the original manuscript.

<Reviewer's Comment #1>

The model is not calibrated. The authors selected a few constant values as the friction values at various locations in the models based on previous studies. Although the authors tried to justify this by saying "in favour of simplicity", I am not convinced that a model without calibration will be of great value.

<Response> (repeating AC1)

We agree with the reviewer that "The tuning of friction values is intentionally kept to minimum in favour of simplicity" is not a good statement. Calibration was indeed an important step during our model setup.

We remove the sentence on Line 156-159 and add more details on calibration:

"The friction of the baseline model was tuned in the wet area (river, estuary, ocean; lower than 1 m above MSL) and on higher grounds (higher than 3 m above MSL) separately. In the wet area, drag coefficients were tested within a range of 0.001-0.01. Commonly accepted default value of 0.0025 gave good error statistics near the landfall site; but the sensitivity is small in the range of 0.001-0.005."

"In the watershed, drag coefficients were tested within a range of 0.01-0.5. The optimal value was chosen based on the High Water Marks (HWMs) comparisons at over two hundred locations, collected by USGS. A small friction value within this range tended to under-predict the elevation at HWMs, and a large value led to over-prediction. Values with the range of 0.02-0.05 gave good error statistics. We chose 0.025 because it gave slightly better results in the Cape Fear River watershed near the landfall."

"Note that this is the parameterization based on the region influenced by Hurricane Florence. Spatially varying parameterization of bottom friction for different systems is an on-going effort as we study more recent hurricanes and operationalize the model along the East Coast and Gulf Coast. However, as we presented in Ye et al. (2020), Zhang et al. (2020) and Huang et al. (2021), the choices described above seem to work fine in general for other systems as well."

[Figure]

(The figure is for the reviewer's reference)

<Reviewer's Comment #2>

Is the NWM model calibrated?

<Response>

Yes, according to Gochis et al. (2018), "all USGS GAGES-II reference basins" and additional gages/basins are included in the calibration and validation; multiple evaluation criteria are applied with emphasis on bias reduction; then parameter sets are regionalized for the whole domain and re-validation.

We are not the developer of the National Water Model (NWM). In our model study, NWM is an external forcing like HYCOM and ERA, which all have uncertainties but are the state-of-the art and the best products available to us. We are open to use any other hydrologic sources to drive our model.

To clarify this, we add the following text after Line 190 of the original manuscript:

"The forcing errors in the magnitude and timing of NWM's peak flow should explain part of the model errors especially in the watershed. For example, we found that replacing the NWM streamflow with gaged flow at USGS Station 02109500 WACCAMAW RIVER AT FREELAND, NC improves the model skill locally. However, this is not cost-effective for our goal of operationalizing this compound flood model along the US East Coast and Gulf Coast. The developers of NWM (Gochis et al., 2018) showed that NWM's model skill was improved by each version update, with 44% of the gauges having bias < +/- 20% in the latest version (NWM v2.0). We will adopt the newest and best NWM version in our ongoing study and operational forecast as soon as it is available. And we are open to using any other hydrologic sources to drive our model."

<Reviewer's Comment #3>

With such detailed 3D modelling of such a large area, what the efficiency of the model is like? E.g. what is the computational time for a flood event lasting for a specific period of time (e.g.3 days)? What is the specs of the computing facility used?

<Response>

We add the information on computational cost after Line 164 in "Section 3. Model description" of the original manuscript:

"Although the model covers a large domain, most of the elements are quasi-2D, making it efficient enough for operational forecast. For the baseline run, the real time to simulation time ratio is 80 with 1440 cores on TACC's Stampede2 and 30 with 480 cores on W&M's Sciclone. Intel Skylake cores with a nominal clock speed of 2.1 GHz were used on both clusters. A 3-day simulation will take about 0.9 hours using 1440 cores or 2.4 hours using 480 cores."

<Reviewer's Comment #4>

The authors reported the average MAE. But often it is the peak error that is important. What is the peak error, when and where did it occur? What is the potential impact of this peak error?

<Response>

Revised as suggested. The revised manuscript reads:

(Line 253) "The peak errors at different stations occur with the storm surge, with a maximum over-prediction of 0.64 m at Spring maid Pier, SC. The overpredicted peak surge can lead to overpredictions in elevation on high water marks (HWMs)."

…

(Line 274) "There is a slight positive bias on near-shore HWMs, corresponding to the over-prediction of peak elevation on coastal stations (Fig. 11)."

…

(Line 371) "Note that the ocean dominance near Wrightsville Beach, NC (NOAA Station 8658163) and Springmaid Pier, SC (NOAA Station 8661070) may be exaggerated (see locations in Fig. 3), considering the overestimated peak elevation there (Fig. 11)

<Reviewer's Comment #5>

Figure 11: The figure shows modelling errors of up to +/-4 meters at various locations. Do the authors have an explanation on the large errors at these locations (apart from just saying calibration can improve model performance)? - The authors did a good job explaining model performance in relation to grid resolution. A similar explanation here will be good.

<Response>

In short, the few large errors are likely the result of similarly **under-resolved bathymetric/topographic features** or the **neglection of drainage system in the urban area**, with minor contributions from other factors such as the uncertainties in the NWM prediction and DEM. We add more explanations into the revised manuscript:

Line 184: "The drainage in urban settings is not included in our model. This may lead to some big errors in the prediction of elevations on high water marks, for example the one large error in the urban area of Fig. 13d. We do have a plan of explicitly accounting for infiltration as volume sinks based on NWM (or other hydrologic models). However, considering the additional uncertainty this would bring, for now we choose to continue improving more important aspects of the model (especially the quality of model grid, which is very likely responsible for the few large errors in Fig. 11c) for operational use."

…

Line 293: "Our tests show that the simulated elevation on the High Water Marks (HWMs) in the watershed is sensitive to: grid resolution, precipitation, river inputs through the land boundary, and bottom friction. Grid resolution and quality is the most important factor, as explained in the last paragraph of Section 4.3. Misrepresentation of flood routing can easily lead to errors of a few meters near some very localized features like ditches, highways etc. How to resolve all important small features efficiently is an on-going research area. The defects in grid quality can lead to large errors that are not likely to be rectified by tuning other parameters. To fix the remaining few large errors away from the landfall site, grid quality should be examined first. The continuous improvement on this model grid is part of an ongoing effort of operationalizing the model along the US East Coast and Gulf Coast, and we will report this in future studies. Afterwards, the inclusion of urban drainage should reduce the occasional large errors there (Fig. 13d). Other factors such as uncertainties in DEM, precipitation and the river flow through land boundary also play minor roles."

…

[Line 300-302] The only remaining large error in the "baseline" occurs in an urban area away from the river, likely due to the building or drainage effects that have not been incorporated in the model (Fig. 13d).

<Reviewer's Comment #6>

What is the return period of flooding at various locations during this event? A comparison on the return period for floods caused with or without compounding effect will give readers a clearer picture of the impact of the compound effect.

We agree with reviewer that a clearer picture of the impact of the compound effect is needed here. US Geological Survey (USGS) provides return period (or annual exceedance probability) of streamflow at selected gages (Feaster et al. 2018). However, many of these gages are at upstream locations of the modeled watershed. On one hand, they are not much affected by the ocean; on the other hand, these streams will be nearly dry without the NWM streamflow injected at the land boundary, making the interpretation of compound effects difficult.

So, we would like to propose an alternative with a more direct measure on the elevation, serving the same purpose of clarifying the compound effect. We propose to calculate the "percent inundated area" or "average inundation depth" inside the North Carolina and South Carolina watersheds and compare this index between the baseline and each sensitivity test.

<Reviewer's Comment #7>

Please see comments on Figure 11.

<Response>

Fig. 14's main purpose is to show the adverse effect of excluding NWM and precipitation. Fig.14 shows more large errors (under-prediction, blue) than in Fig. 11 (baseline), because Fig.14's has no stream flow from NWM or direct precipitation.

We add more explanations in the 2nd paragraph of Section 5 to convey the idea more clearly:

"Not surprisingly, without rivers and precipitation, watershed is mostly dry as the storm surge cannot propagate over steep terrains (Fig. 12). As a result, the predicted HWMs are biased too low (Fig. 14) as the steep topography quickly damped out any surges brought in by the ocean. *This leads to systematically underpredictions in the watershed and a 64% increase in MAE compared to the baseline (Fig. 11)*. … *Note that there is no apparent deterioration of model skill at the near-shore HWMs, because those locations are predominately affected by the oceanic processes.*"

<Reviewer's Minor Comment #1>

Figure 15. I understand this figure is used to show the impact from different compound flood driver. It is difficult to interpret the results. The caption can include some accompany text on how the figure can be interpreted (what the proportion values imply).

<Response>

Following the reviewer's suggestion, we will change Fig 15's caption to:

Fig. 15. Regional map showing the spatially-varying importance of each forcing factor: (a) ocean force ("Ocean" in Table 1); (b) river force ("NWM" in Table 1); and (c) precipitation ("Rain" in Table 1)." The value is the proportion of a factor's individual "disturbance" (see definition in Section 5) to the sum of the disturbance from all factors. The colors from blue to red represent increasing importance of a factor at a specific location.

We also add texts to the labels inside the figure:

[Figure]

<References>

Feaster, T.D., Weaver, J.C., Gotvald, A.J. and Kolb, K.R., 2018. Preliminary peak stage and streamflow data for selected US Geological Survey streamgaging stations in North and South Carolina for flooding following Hurricane Florence, September 2018 (No. 2018-1172). US Geological Survey.

Huang, W., Ye, F., Zhang, Y.J., Park, K., Du, J., Moghimi, S., Myers, E., Pe'eri, S., Calzada, J.R., Yu, H.C. and Nunez, K., 2021. Compounding factors for extreme flooding around Galveston Bay during Hurricane Harvey. *Ocean Modelling*, *158*, p.101735.

Gochis, D.J., Cosgrove, B., Dugger, A.L., Karsten, L., Sampson, K.M., McCreight, J.L., Flowers, T., Clark, E.P., Vukicevic, T., Salas, F. and FitzGerald, K., 2018, December. Multi-variate evaluation of the NOAA National Water Model. In *AGU Fall Meeting Abstracts* (Vol. 2018, pp. H33G-01).

Ye, F., Zhang, Y.J., Yu, H., Sun, W., Moghimi, S., Myers, E., Nunez, K., Zhang, R., Wang, H.V., Roland, A. and Martins, K., 2020. Simulating storm surge and compound flooding events with a creek-to-ocean model: Importance of baroclinic effects. *Ocean Modelling*, *145*, p.101526.

Zhang, Y.J., Ye, F., Yu, H., Sun, W., Moghimi, S., Myers, E., Nunez, K., Zhang, R., Wang, H., Roland, A. and Du, J., 2020. Simulating compound flooding events in a hurricane. *Ocean Dynamics*, *70*(5), pp.621-640.

---

## Author Comment (AC3) · 15 Mar 2021

Dear Reviewer,

Thanks very much for your positive feedback and very useful comments. We fix the problems and address your comments as listed below. Your original comments are copied for your reference. Following the journal's publishing guideline, the revised manuscript is not uploaded at this time, so all line numbers and figure numbers are based on the original manuscript.

<Reviewer's Comment #1>

The authors suggest that a 3D baroclinic model is necessary to accurately capture the water level response. However, I suspect that utilizing such a complex model (plus the very large mesh size) would require a very large computational expense. The authors do not present results from using a more simplified approach (2D depth-averaged for example), so we don't know whether the additional complexity is actually needed to capture the water level response. The manuscript would benefit from some discussion about the trade offs between model complexity and computational burden, as well as some details about how long each simulation takes to run.
<Response>

We agree with the reviewer and add more texts after Line 105:

 "The trade-off between 2D and 3D must be carefully weighed, especially for operationalization. In short, the advantage of 2D is the speed (about 80 times faster than 3D baroclinic) and the simplicity of the set up. The disadvantage of 2D is that it misses the effects from baroclinic processes that may become important at certain time and locations. The baroclinic effects during the adjustment phase after Hurricane Irene (2011) are discussed in details in Ye et al. (2020), using a similar model setup as the one used here. Even though different setups (2D, 3D barotropic, and 3D baroclinic) were tuned to the best possible skill, the 3D baroclinic setup was shown to better capture the post-storm adjustment phase. In addition, during the ongoing effort to operationalize the model, we found that including 3D processes greatly reduces the need for bottom friction parameterization at some coastal locations."

The figure below shows the model-data comparison for 2D and 3D with a similar model setup as the current paper. Although it is possible to improve the 2D skill by locally tuning bottom friction, the 3D model achieved good skill without any local tuning. We are looking into the momentum budget at this location and will report the findings in a future publication.

[Figure]

We also add the information on computational cost after Line 164 in "Section 3. Model description" of the original manuscript:

"Although the model covers a large domain, most of the elements are quasi-2D, making it efficient enough for operational forecast. For the baseline run, the real time to simulation time ratio is 80 with 1440 cores on TACC's Stampede2 and 30 with 480 cores on W&M's Sciclone. Intel Skylake cores with a nominal clock

speed of 2.1 GHz were used on both clusters. A 3-day simulation will take about 0.9 hours using 1440 cores or 2.4 hours using 480 cores."

<Reviewer's Comment #2>

In my experience, the NCEI CUDEM does not capture the bathymetry of coastal streams well, and tends to significantly underestimate the channel depth. Did the authors make any corrections/modifications to the raw DEM to better represent coastal streams?

<Response>

No, we did not do any modifications on the NCEI CUDEM. We also noticed some underestimation of channel depth, especially in the South Carolina watersheds, which may have contributed to the errors on HWMs in Fig. 11c.

Our collaborators at NOAA (including those in the author list) is working on introducing improved DEMs to our model, including some DEMs not yet open to the public. Sometimes we manually correct DEMs to make channels continuous based on navigation charts, imagery, and our best judgement; but we have not done so in the current study.

We will add this information on Line 137 of the original manuscript, so that the revised manuscript reads:

"Customary of all SCHISM applications, no manipulation or smoothing of bathymetry was done in the computational grid after interpolation of the depths from DEMs (including steep slopes in the Caribbean and all shipping channels). From our experience, CUDEM may underestimate the depth of coastal streams (e.g., in the South Carolina watersheds), which is a potential error source of our model."

<Reviewer's Comment #3>

There is no mention of infiltration in this manuscript, which is an important factor controlling the pluvial flood dynamics. Did the authors completely neglect infiltration in this work? And if so, there should be some discussion/justification about why infiltration was not accounted for. Since much of the underlying soil in the NC region is sand, I expect infiltration losses would be non-negligible and neglecting infiltration could cause over-estimation of the rainfall-induced flooding.

<Response>

Yes, we neglected infiltration in this work. We agree that infiltration may be an important process. But in the case of Hurricane Florence induced flooding, we expect the effect of infiltration to be minor.

The revised manuscript reads:

[Line 184] As a model limitation, infiltration is neglected in this work. In the case of Hurricane Florence induced flooding, we expect the effect of infiltration to be minor. According to the NOAA's weather map (https://www.wpc.ncep.noaa.gov/dailywxmap/), there was continuous rainfall along the US east coast from Sep 11, 2018 to the date of Florence's landfall (Sep 14, 2018), so the infiltration capacity of the soil was already reduced. Moreover, "wet" storms like Hurricane Florence (2018) and Hurricane Harvey (2017) tend to dump large amount of rain fall at a location for days because of the slow movement of the storm, so most of the rainfall is on saturated soil. The drainage in urban settings is not included in our model. This may

lead to some big errors in the prediction of elevations on high water marks; for example, the one large error in the urban area of Fig. 13d. We do have a plan of explicitly accounting for infiltration as volume sinks based on NWM (or other hydrologic models). However, considering the additional uncertainty this would bring, for now we choose to continue improving more important aspects of the model (especially the quality of model grid, which is likely responsible for most of the large errors in Fig. 11c) for operational use.

< Reviewer's Specific Comment #1>

Line 106: What is meant by "bona fide" in this context? Please clarify.

<Response>

We delete these two words and clearly identify the merit of the current model:

As we shall see, this model solves the physical processes from the watershed to the ocean with the same set of governing equations, qualifying for Santiago-Collazo et al. (2019)'s definition of a fully-coupled compound surge and flood model.

< Reviewer's Specific Comment #2>

Figure 3: I am a bit confused by figures 3b and 3c. The box showing the location of figure 3c indicates that 3c is north of Albermale Sound. However, when I look at Figure 3c it appears to be depicting the Cape Fear River Estuary, which is at the southern tip of NC. Please confirm/update the actual location of Fig 3c.

<Response>

Thanks for pointing this out. We have corrected the location of Fig. 3c:

[Figure]

< Reviewer's Specific Comment #3>

Table 1: The difference between the No_NWM_precip and Ocean scenario is not clear to me. It seems they both do not include river or precip forcing?

<Response>

 "No_NWM_precip" and "Ocean scenario" are identical. We will only use "Ocean scenario" in the revised manuscript. Thanks for catching this issue.

< Reviewer's Specific Comment #4>

Fig 9: It is difficult to distinguish the differences between the model runs, especially for figures 9a-d. I suggest the authors shorten the depicted time window to only show 1-2 days before landfall through 4-5 days after landfall for a-d. This will allow a reader to see more clearly the differences between the scenarios and the comparison to the observed water levels.

<Response>

Another reviewer also asked for changes on this figure. To accommodate both reviewers' request, we will shorten the time window and also add sub-tidal comparisons in Fig 9b (see AC3).

**<References>**

Santiago-Collazo, F.L., Bilskie, M.V., Hagen, S.C.: A comprehensive review of compound inundation models in low-gradient coastal watersheds. Environ. Model. Software 119:166–181. https://doi.org/10.1016/j.envsoft.2019.06.002, 2019.

Ye, F., Zhang, Y.J., Yu, H., Sun, W., Moghimi, S., Myers, E., Nunez, K., Zhang, R., Wang, H.V., Roland, A. and Martins, K., 2020. Simulating storm surge and compound flooding events with a creek-to-ocean model: Importance of baroclinic effects. *Ocean Modelling*, *145*, p.101526.

Zhang, Y.J., Ye, F., Yu, H., Sun, W., Moghimi, S., Myers, E., Nunez, K., Zhang, R., Wang, H., Roland, A. and Du, J., 2020. Simulating compound flooding events in a hurricane. Ocean Dynamics, 70(5), pp.621-640.

---

## Author Comment (AC4) · 15 Mar 2021

Dear Reviewer,

Thanks very much for taking the time to help us improve the manuscript. We have copied your questions/comments below for your reference and respond to each. Following the journal's publishing guideline, the revised manuscript is not uploaded at this time, so all line numbers and figure numbers are based on the original manuscript.

**<Reviewer's general comment>**

To me the novel aspect of this work is having very much detailed representation of hydrology (via NWM based on WRF), river morphology and coastal ocean dynamics (via a 3D baroclinic model). The manuscript is well written and can be of interest of community, thus I suggest publication in NHESS, however after a major revision. The major issue to me is that, despite the efforts of authors to calibrate/validate their model, on the hydrologic fluvial side the model does not seem well calibrated, and the significant error in NWM prediction (both timing and magnitude of peak) can pose a significant error to the overall estimates. Also, not clear to me how the pluvial flooding processes (from direct rainfall) is treated here. More detailed comments below.

**<Response to general comment>**

We need to clarify a possible misunderstanding on our model. NWM (based on WRFHydro) is only used as a boundary condition like HYCOM and ERA, which all have uncertainties but are the state-of-the-art. NWM segments inside the model domain are only used to guide mesh generation, because these segments correspond to thalwegs in the DEM.

We are not the developers of NWM, so the quality of NWM prediction is out of our control. It is the best hydrologic products available to us for this regional-scale study; we are fully open to other hydrologic models.

The merit of this work is being an effort toward a fully-coupled compound inland flood and ocean related surge model. Here "fully-coupled" means the model solves its included processes with the same set of governing equations (Santiago-Collazo et al., 2019). Specifically, we include enough watershed region in our model domain so that the interaction between pluvial/fluvial processes and estuarine/oceanic processes can be simulated directly, without the need for "coupling" or "link" among different types of models. In the watershed, the movement of water (flood routing) is purely controlled by physics (governing equation), just like how the estuarine and the ocean circulation are simulated. This poses great challenge on the model's robustness and efficiency.

We probably did not make this clear enough in the first paragraph of "3.2 Coupling with NWM". In the revised manuscript, we will add more details (see responses to Reviewer's Major Comment #1 and Reviewer's Major Comment #2 below) and convey a clearer message on NWM's role in our model.

**<Reviewer's Major Comment #1>**

Hydrologic processes are not well represented here. River flow hydrograph estimates from NWM are significantly different than the observed flow at USGS gauges (see Figure 6). About 2-3 days of lag in peak

flow estimation and up to 100% error in peak flow magnitude can pose a significant error in overall inundation and coastal water level dynamics if propagated through the system. Timing (and magnitude) of peak runoff plays a significant role in extreme water level dynamics in freshwater-influenced coastal systems and getting these hydrologic characteristics right plays a major role in accurate estimation of extent/intensity of compound flooding. This might partially explain the wide range of difference (up to 4 m) between simulated and estimated high water marks (Figure 11), with a considerable number of points having estimation error greater than 1m (Figure 11e).

**<Response>**

We agree with the reviewer that part of our model errors can be attributed to the errors from NWM. We add the following text after Line 190 of the original manuscript to explain our choice:

"The observation indicates the peak streamflow occurs about 7 days after the landfall, which is the time it takes for the rainfall induced flood to reach the coastal rivers. Note that there is typically a time lag of 1-2 days between the peak flow in NWM and the gaged flow. The forcing errors in the magnitude and timing of NWM's peak flow should explain part of the model errors especially in the watershed. For example, we found that replacing the NWM streamflow with gaged flow at USGS Station 02109500 WACCAMAW RIVER AT FREELAND, NC improves the model skill locally. However, this is not cost-effective for our goal of operationalizing this compound flood model along the US East Coast and Gulf Coast. The developers of NWM (Gochis et al., 2018) showed that NWM's model skill was improved by each version update, with 44% of the gauges having bias < +/- 20% in the latest version (NWM v2.0). We will adopt the newest and best NWM version in our ongoing study and operational forecast as soon as it is available. And we are open to using any other hydrologic sources to drive our model."

**<Reviewer's Major Comment #2>**

Could not find any information on how the pluvial processes are handled in the model. Rain-on grid modeling (L174) is a good contribution to the field which I acknowledge, but how the underlying processes and variables that contribute to the runoff generation are accounted for is vague. How infiltration capacity is accounted for? How drainage infrastructure contribution is accounted for (in urban settings)? How the generated runoff is routed between pixels? This is even more important when the results suggest that flooding in a significant portion of coastal land in dominated by "precipitation only" (Figure 15c).

**<Response>**

We removed the original text on Line 171-173. We state that the neglection of infiltration and drainage is a limitation of the model that warrants further work. The relevant texts in the revised manuscript read:

[Revised Line 106 of the original manuscript] This model solves the physical processes from the watershed to the ocean with the same set of governing equations, qualifying for Santiago-Collazo et al. (2019)'s definition of a fully-coupled compound surge and flood model.

•••

[Revised Line 171-174 of the original manuscript] Over the model domain, streamflow injected at the land boundary and precipitation are directly handled by the hydrodynamic core of the ocean model (SCHISM). This fully coupled configuration is rare in the existing compound flooding simulations (Santiago-Collazo et al., 2019). To ensure the accuracy and robustness of SCHISM in simulating hydrological and hydraulic processes, we examined the model's performance in both lab-scale and field-scale tests in a previous study

(Section 2.2 and 2.3 in Zhang et al. (2020)) and applied the model in the Delaware Bay watershed including part of the Delaware River up to 40 meters above the NAVD88 datum with a hydraulic jump (Fig. 14 in Zhang et al. (2020))."

•••

[Line 184] As a model limitation, infiltration is neglected in this work. In the case of Hurricane Florence induced flooding, we expect the effect of infiltration to be minor. According to the NOAA's weather map (https://www.wpc.ncep.noaa.gov/dailywxmap/), there was continuous rainfall along the US east coast from Sep 11, 2018 to the date of Florence's landfall (Sep 14, 2018), so the infiltration capacity of the soil was already reduced. Moreover, "wet" storms like Hurricane Florence (2018) and Hurricane Harvey (2017) tend to dump large amount of rain fall at a location for days because of the slow movement of the storm, so most of the rainfall is on saturated soil. The drainage in urban settings is not included in our model. This may lead to some big errors in the prediction of elevations on high water marks; for example, the one large error in the urban area of Fig. 13d. We do have a plan of explicitly accounting for infiltration as volume sinks based on NWM (or other hydrologic models). However, considering the additional uncertainty this would bring, for now we choose to continue improving more important aspects of the model (especially the quality of model grid, which is likely responsible for most of the large errors in Fig. 11c) for operational use.

•••

[Line 300-302] The only remaining large error in the "baseline" occurs in an urban area away from the river, likely due to the building or drainage effects that have not been incorporated in the model (Fig. 13d).

< Reviewer's Minor Comment #1>

Figure 1: red text (in the left panels) is not readable in print. I'd also add some space between rows in the legend.

<Response>

Revised as suggested:

---

## Author Comment (AC5) · 15 Mar 2021

We found a minor problem with Fig. 13d and fixed it in the new Fig. 13 below. Specifically, a different run other than the baseline model was used to plot Fig. 13d in the original manuscript, which has been corrected. This does not change the conclusion that resolving small-scale flow routing features greatly improves the prediction of HWM elevation. The "Burnt Mill Creek" was mis-labeled as "Northeast Cape Fear River" in the original manuscript, which has been corrected. Lake Greenfield and more labels were also added for better illustration.

Fig. 13: Importance of resolving small-scale features on the order of a few meters in the watershed, illustrated by a comparison between a preliminary setup (a, b) and the

baseline setup (c, d). To better resolve the Burnt Mill Creek, NC, more SMS feature arcs (cyan lines in (d)) are used in the baseline setup than in the preliminary setup (cyan lines in (b)), significantly reducing the HWM errors. See Fig. 2 for the location of this locally zoomed-in region. The base maps in (b) and (d) are provided by ESRI.
* * *
[Figure]

Fig. 1.

---

## Author Response (AR1)

**Authors' Reply**

Dear reviewers,

Thank you for your time and effort in helping us improve the manuscript.

As suggested by the editor, we repeat the previously uploaded authors' comments (AC) with slight changes and upload the revised manuscript. A "track-change" version of the revised manuscript is also attached at the end of this document.

If you have not gone through the previous ACs, you can just read this document, in which we address all your questions and comments. Your original comments are repeated for your reference. Line numbers from the original manuscript (OM) and the revise manuscript (RM) are both provided.

**Reply to Reviewer #1**

<Reviewer's Comment #1>

The model is not calibrated. The authors selected a few constant values as the friction values at various locations in the models based on previous studies. Although the authors tried to justify this by saying "in favour of simplicity", I am not convinced that a model without calibration will be of great value.

<Response>

We agree with the reviewer that it is not a good statement to say "The tuning of friction values is intentionally kept to minimum in favour of simplicity". Calibration was indeed an important step during our model setup.

We remove the statement on Line 156-159 (OM) and add more details on calibration (Line 168, RM):

"The friction of the baseline model was tuned in the wet area (river, estuary, ocean; lower than 1 m, NAVD88) and on higher grounds (higher than 3 m, NAVD88) separately. In the wet area, drag coefficients within a range of 0.001-0.01 were tested. Commonly accepted default value of 0.0025 gave good error statistics near the landfall site; but the sensitivity is small in the range of 0.001-0.005."

"In the watershed, drag coefficients within a range of 0.01-0.5 were tested. The optimal value was chosen based on the High Water Marks (HWMs) comparisons at 276 locations, collected by USGS. A small friction value within this range tended to under-predict the elevation at HWMs, and a large value led to over-prediction. Values within the range of 0.02-0.05 gave good error statistics. We chose 0.025 because it gave slightly better results in the Cape Fear River watershed near the landfall."

"Note that this is the parameterization based on the region influenced by Hurricane Florence. Spatially varying parameterization of bottom friction for different systems is an on-going effort as we study more recent hurricanes and operationalize the model along the East Coast and Gulf Coast. However, as we presented in Ye et al. (2020), Zhang et al. (2020) and Huang et al. (2021), the choices described above seem to work fine in general for other systems as well."

[Figure]

(The figure is for the reviewer's reference)

<Reviewer's Comment #2>

Is the NWM model calibrated?

<Response>

Yes, according to Gochis et al. (2018), "all USGS GAGES-II reference basins" and additional gages/basins are included in the calibration and validation; multiple evaluation criteria are applied with emphasis on bias reduction; then parameter sets are regionalized for the whole domain and re-validation.

We are NOT the developer of the National Water Model (NWM). In our model study, NWM is an external forcing just like HYCOM and ERA, which all have uncertainties but are the state-of-the art and the best products available to us. We are open to use any other hydrologic sources to drive our model.

To clarify this, we add the following text after Line 190 of the original manuscript (Line 232, RM):

"The forcing errors in the magnitude and timing of NWM's peak flow should explain part of the model errors especially in the watershed. For example, we found that replacing the NWM streamflow with the gauged flow at USGS Station 02109500 (Waccamaw River at Freeland, NC) improves the model skill locally. However, this is not cost-effective for our goal of operationalizing this compound flood model along the US East Coast and Gulf Coast. The developers of NWM (Gochis et al., 2018) showed that NWM's model skill was improved by each version update, with 44% of the gauges having bias < 20% in the latest version (NWM v2.0). We will adopt the newest and best NWM version in our ongoing study and operational forecast as soon as it is available. And we are open to using any other hydrologic sources to drive our model."

<Reviewer's Comment #3>

With such detailed 3D modelling of such a large area, what the efficiency of the model is like? E.g. what is the computational time for a flood event lasting for a specific period of time (e.g.3 days)? What is the specs of the computing facility used?

<Response>

We add the information on computational cost (Line 164, OM; Line 186, RM) and in "Section 3. Model description" of the original manuscript:

"Although the model covers a large domain, most of the elements (those in the watersheds) are quasi-2D, making it efficient enough for operational forecast. For the baseline run, the real time to simulation time ratio is 80 with 1440 cores on TACC's Stampede2 cluster and 30 with 480 cores on W&M's SciClone cluster. Intel Skylake cores with a nominal clock speed of 2.1 GHz were used on both clusters. This means a 3-day simulation (typical forecast duration) will take about 0.9 hours using 1440 cores or 2.4 hours using 480 cores."

<Reviewer's Comment #4>

The authors reported the average MAE. But often it is the peak error that is important. What is the peak error, when and where did it occur? What is the potential impact of this peak error?

<Response>

Revised as suggested. The revised manuscript reads:

(Line 253, OM; Line 307, RM) "The peak errors at different stations occur around the storm surge, with a maximum over-prediction of 0.64 m at Spring maid Pier, SC. The overpredicted peak surge can lead to overpredictions in elevation on coastal high water marks (HWMs). In addition, there is a maximum under-prediction of 0.66 m for the set down at Hatteras, NC, mainly due to the mismatch in the set down timing. The uncertainties in wind forcing may be the main cause of the error, which is predominantly from subtidal signals. The grid quality near the barrier islands may also contribute to the error."

…

(Line 274, OM; Line 337, RM) "There is a slight positive bias on near-shore HWMs (Fig. 11), corresponding to the over-prediction of peak elevation at coastal stations (Fig. 9)."

…

(Line 371, OM; Line 450, RM) "Note that the ocean dominance near Wrightsville Beach, NC (NOAA Station 8658163) and Springmaid Pier, SC (NOAA Station 8661070) may be exaggerated, considering the overestimated peak elevation there (Fig. 9)."

<Reviewer's Comment #5>

Figure 11: The figure shows modelling errors of up to +/-4 meters at various locations. Do the authors have an explanation on the large errors at these locations (apart from just saying calibration can improve model

performance)? - The authors did a good job explaining model performance in relation to grid resolution. A similar explanation here will be good.

<Response>

In short, the few large errors are likely the result of similarly **under-resolved bathymetric/topographic features** or the **neglection of drainage system in the urban area**, with minor contributions from other factors such as the uncertainties in the NWM prediction and DEM. We add more explanations into the revised manuscript:

[Line 184, OM; Line 221, RM]: "As another model limitation, the drainage in urban settings is not included in our model. This may have led to some occasional big errors in the predicted elevation on high water marks, for example the one large error in the urban area in Fig. 13d. We do have a plan of explicitly accounting for infiltration and drainage as volume sinks based on NWM (or other hydrologic models). However, considering the additional uncertainty this would bring, for now we choose to continue improving more important aspects of the model for operational use; the focus is on the quality of model grid, which is very likely responsible for most of the existing large errors."

…

[Line 293, OM; Line 357, RM]: "Our tests show that the simulated elevation on the High Water Marks (HWMs) in the watershed is sensitive to: grid resolution, precipitation, river inputs through the land boundary, and bottom friction. Grid resolution and quality is the most important factor, as explained in the last paragraph of Section 4.3. Misrepresenting flood pathways can easily lead to errors of a few meters near some very localized features like ditches, highways etc. … The defects in grid quality can lead to large errors that are not likely to be rectified by tuning other parameters. To fix the remaining few large errors away from the landfall site, grid quality should be examined first. The continuous improvement on this model grid is part of an ongoing effort of operationalizing the model along the US East Coast and Gulf Coast, and we will report this in future studies. … Afterwards, the inclusion of urban drainage should reduce the occasional large errors there (Fig. 13d). Other factors such as uncertainties in DEM, precipitation and the river flow through land boundary also play minor roles."

…

[Line 300-302, OM; Line 364, RM] The only remaining large error in the "baseline" occurs in an urban area away from the river, likely due to the building or drainage effects that have not been incorporated in the model (Fig. 13d).

<Reviewer's Comment #6>

What is the return period of flooding at various locations during this event? A comparison on the return period for floods caused with or without compounding effect will give readers a clearer picture of the impact of the compound effect.

We agree with reviewer that a clearer picture of the impact of the compound effect is needed here. US Geological Survey (USGS) provides return period (or annual exceedance probability) of streamflow at selected gages (Feaster et al. 2018). However, many of these gages are at upstream locations of the modeled watershed. On one hand, they are not much affected by the ocean; on the other hand, these streams will be nearly dry without the NWM streamflow injected at the land boundary, making the interpretation of compound effects difficult.

So, we would like to propose an alternative with a more direct measure on the elevation, serving the same purpose of clarifying the compound effect. We propose to calculate the "percent inundated area" or "average inundation depth" inside the North Carolina and South Carolina watersheds and compare this index between the baseline and each sensitivity test.

In the revised manuscript, we added the following text and table on Line 388:

"As an overview, the conditions of maximum inundation extent from all scenarios are listed in Table 2. To facilitate the comparison of inundated area, a practical value (1 foot) on the same order of the mean inundation depth is used as a threshold. For the two indices shown in Table 2, the baseline values are always larger than those from a single sensitivity test. This confirms the existence of compound regions in the two states (North Carolina and South Carolina) during the hurricane. More details of each forcing's effect and the compound effects are discussed below."

**Table 2: Overview of the maximum inundation extent in South Carolina and North Carolina watersheds (above the NAVD88 datum) during the simulation period.**

| Name | Percentage of inundated area with water depth > 0.305 m (1 foot) | Spatially average inundation depth (m) |
|---|---|---|
| Baseline | 46.7% | 0.61 |
| Ocean | 12.7% | 0.12 |
| River | 17.4% | 0.31 |
| Rain | 34.4% | 0.36 |

<Reviewer's Comment #7>

Please see comments on Figure 11.

<Response>

Fig. 14's main purpose is to show the adverse effect of excluding NWM and precipitation. Fig.14 shows more large errors (under-prediction, blue) than in Fig. 11 (baseline), because Fig.14's has no stream flow from NWM or direct precipitation.

We add more explanations in the 2nd paragraph of Section 5 to convey the idea more clearly:

"Not surprisingly, without rivers and precipitation, watershed is mostly dry as the storm surge cannot propagate over steep terrains (Fig. 12). As a result, the predicted HWMs are biased too low (Fig. 14) as the steep topography quickly damped out any surges brought in by the ocean. *This leads to systematically underpredictions in the watershed and a 64% increase in MAE compared to the baseline (Fig. 11). … Note that there is no apparent deterioration of model skill on the near-shore HWMs, because those locations are predominately affected by oceanic processes.*"

<Reviewer's Minor Comment #1>

Figure 15. I understand this figure is used to show the impact from different compound flood driver. It is difficult to interpret the results. The caption can include some accompany text on how the figure can be interpreted (what the proportion values imply).

<Response>

Following the reviewer's suggestion, we change Fig 15's caption to:

Fig. 15. Regional map showing the spatially varying importance of each forcing factor: (a) ocean force ("Ocean" in Table 1); (b) river force ("NWM" in Table 1); and (c) precipitation ("Rain" in Table 1). The value is the proportion of a factor's individual "disturbance" (see definition in Section 5) to the sum of the disturbance from all factors. The colors from blue to red represent increasing importance of a factor at a specific location.

We also add texts to the labels inside the figure:

[Figure]

<References>

Feaster, T.D., Weaver, J.C., Gotvald, A.J. and Kolb, K.R., 2018. Preliminary peak stage and streamflow data for selected US Geological Survey streamgaging stations in North and South Carolina for flooding following Hurricane Florence, September 2018 (No. 2018-1172). US Geological Survey.

Huang, W., Ye, F., Zhang, Y.J., Park, K., Du, J., Moghimi, S., Myers, E., Pe'eri, S., Calzada, J.R., Yu, H.C. and Nunez, K., 2021. Compounding factors for extreme flooding around Galveston Bay during Hurricane Harvey. *Ocean Modelling*, *158*, p.101735.

Gochis, D.J., Cosgrove, B., Dugger, A.L., Karsten, L., Sampson, K.M., McCreight, J.L., Flowers, T., Clark, E.P., Vukicevic, T., Salas, F. and FitzGerald, K., 2018, December. Multi-variate evaluation of the NOAA National Water Model. In *AGU Fall Meeting Abstracts* (Vol. 2018, pp. H33G-01).

Ye, F., Zhang, Y.J., Yu, H., Sun, W., Moghimi, S., Myers, E., Nunez, K., Zhang, R., Wang, H.V., Roland, A. and Martins, K., 2020. Simulating storm surge and compound flooding events with a creek-to-ocean model: Importance of baroclinic effects. *Ocean Modelling*, *145*, p.101526.

Zhang, Y.J., Ye, F., Yu, H., Sun, W., Moghimi, S., Myers, E., Nunez, K., Zhang, R., Wang, H., Roland, A. and Du, J., 2020. Simulating compound flooding events in a hurricane. *Ocean Dynamics*, *70*(5), pp.621-640.

**End of Reply to Reviewer #1**

**Reply to Reviewer #2**

<Reviewer's Comment #1>

The authors suggest that a 3D baroclinic model is necessary to accurately capture the water level response. However, I suspect that utilizing such a complex model (plus the very large mesh size) would require a very large computational expense. The authors do not present results from using a more simplified approach (2D depth-averaged for example), so we don't know whether the additional complexity is actually needed to capture the water level response. The manuscript would benefit from some discussion about the trade offs between model complexity and computational burden, as well as some details about how long each simulation takes to run.

<Response>

We agree with the reviewer and add more texts (before Line 50, OM; Line 49, RM)

 "*The trade-off between 2D and 3D was carefully weighed with the goal of operationalization in mind before the latter was chosen. In short, the advantage of 2D is the speed (about 80 times faster than its 3D baroclinic counterpart) and the simplicity of the set up; the disadvantage is that it misses baroclinic effects and 3D processes that may become important at certain times and locations. For example, the baroclinic effects during the adjustment phase after Hurricane Irene (2011) are discussed in detail in Ye et al. (2020), using a similar model setup as the one used here. Even though different setups (2D, 3D barotropic, and 3D baroclinic) were tuned to their best possible skills, the 3D baroclinic setup was shown to better capture the total elevation during the post-storm adjustment phase. In addition, during the ongoing effort to operationalize the model, we found that including 3D processes greatly simplified the bottom friction parameterization at some coastal locations (e.g., NOAA Station 8447930 at Woods Hole, MA; Huang et al., submitted).* A 3D model can also produce relevant 3D variables (e.g., 3D velocity and tracer concentration) that are important for safe navigation and ecosystem health. The 3D model presented in this paper is efficient enough for operational forecasts (see Section 3.1), which are being set up at NOAA (National Oceanic and Atmospheric Administration)."

The figure below shows the model-data comparison for 2D and 3D with a similar model setup as the current paper. Although it is possible to improve the 2D skill by locally tuning bottom friction, the 3D model achieved good skill without any local tuning. The momentum budget around this location shows that the baroclinic pressure gradient is comparable to the barotropic pressure gradient.

[Figure]

We also add the information on computational cost at the end of "Section 3.1 Model setups":

"Although the model covers a large domain, most of the elements (those in the watersheds) are quasi-2D, making it efficient enough for operational forecast. For the baseline run, the real time to simulation time ratio is 80 with 1440 cores on TACC's Stampede2 cluster and 30 with 480 cores on W&M's Sciclone cluster. Intel Skylake cores with a nominal clock speed of 2.1 GHz were used on both clusters. A 3-day (typical forecast duration) simulation will take about 0.9 hours using 1440 cores or 2.4 hours using 480 cores."

<Reviewer's Comment #2>

In my experience, the NCEI CUDEM does not capture the bathymetry of coastal streams well, and tends to significantly underestimate the channel depth. Did the authors make any corrections/modifications to the raw DEM to better represent coastal streams?

<Response>

No, we did not do any modifications on the NCEI CUDEM. We also noticed some underestimation of channel depth, especially in the South Carolina watersheds, which may have contributed to the errors on HWMs in Fig. 11c.

Our collaborators at NOAA (including those in the author list) is working on introducing improved DEMs to our model, including some DEMs not yet open to the public. Sometimes we manually correct DEMs to make channels continuous based on navigation charts, imagery, and our best judgement; but we have not done so in the current study.

We add this information on Line 137 of the original manuscript (Line 148, RM), so that the revised manuscript reads:

"Customary of all SCHISM applications, no manipulation or smoothing of bathymetry was done in the computational grid after interpolation of the depths from DEMs (including steep slopes in the Caribbean and all shipping channels). *From our experience, CUDEM may underestimate the depth of coastal streams (e.g., in the South Carolina watersheds), which is a potential error source of our model.*"

<Reviewer's Comment #3>

There is no mention of infiltration in this manuscript, which is an important factor controlling the pluvial flood dynamics. Did the authors completely neglect infiltration in this work? And if so, there should be some discussion/justification about why infiltration was not accounted for. Since much of the underlying soil in the NC region is sand, I expect infiltration losses would be non-negligible and neglecting infiltration could cause over-estimation of the rainfall-induced flooding.

<Response>

Yes, we neglected infiltration in this work. We agree that infiltration may be an important process. But in the case of Hurricane Florence induced flooding, we expect the effect of infiltration to be minor.

The revised manuscript reads:

[Line 184, OM; Line 215, RM] As a model limitation, infiltration is neglected in this work. In the case of Hurricane Florence induced flooding, we expect the effect of infiltration to be minor. According to NOAA's

weather map (https://www.wpc.ncep.noaa.gov/dailywxmap/), there was continuous rainfall along the US east coast from Sep 11, 2018 to the date of Florence's landfall (Sep 14, 2018), so the infiltration capacity of the soil was already reduced. Moreover, "wet" storms like Hurricane Florence (2018) and Hurricane Harvey (2017) tend to dump large amount of rain fall at a location for days because of the slow movement of the storm, so most of the rainfall is on saturated soil. As another model limitation, the drainage in urban settings is not included in our model. This may have led to some occasional big errors in the predicted elevation on high water marks, for example the one large error in the urban area in Fig. 13d. We do have a plan of explicitly accounting for infiltration and drainage as volume sinks based on NWM (or other hydrologic models). However, considering the additional uncertainty this would bring, for now we choose to continue improving more important aspects of the model for operational use; the focus is on the quality of model grid, which is very likely responsible for most of the existing large errors.

< Reviewer's Specific Comment #1>

Line 106: What is meant by "bona fide" in this context? Please clarify.

<Response>

We delete these two words and clearly identify the merit of the current model (Line 106, OM; Line 114, RM):

As we shall see, this model solves the physical processes from the watershed to the ocean with the same set of governing equations, qualifying for Santiago-Collazo et al. (2019)'s definition of a fully-coupled compound surge and flood model.

< Reviewer's Specific Comment #2>

Figure 3: I am a bit confused by figures 3b and 3c. The box showing the location of figure 3c indicates that 3c is north of Albermale Sound. However, when I look at Figure 3c it appears to be depicting the Cape Fear River Estuary, which is at the southern tip of NC. Please confirm/update the actual location of Fig 3c.

<Response>

Thanks for pointing this out. We have corrected the location of Fig. 3c:

[Figure]

< Reviewer's Specific Comment #3>

Table 1: The difference between the No_NWM_precip and Ocean scenario is not clear to me. It seems they both do not include river or precip forcing?

<Response>

"No_NWM_precip" and "Ocean scenario" are identical. We only use "Ocean scenario" in the revised manuscript. Thanks for catching this issue.

< Reviewer's Specific Comment #4>

Fig 9: It is difficult to distinguish the differences between the model runs, especially for figures 9a-d. I suggest the authors shorten the depicted time window to only show 1-2 days before landfall through 4-5 days after landfall for a-d. This will allow a reader to see more clearly the differences between the scenarios and the comparison to the observed water levels.

<Response>

To improve Fig. 9, we shorten the time window, change the line style of the "Baseline_Wave", and also add sub-tidal comparisons:

[Figure]

Fig. 9: Comparison of elevation at six NOAA gauges: (left) total elevation; (right) subtidal elevation. Also included are results from two sensitivity runs ("Baseline_Wave" and "No_NWM_precipOcean"; cf. Table 1). The stations in (a)-(d) are exposed to the ocean, showing small differences among scenarios; the two

stations in (e) and (f) are on the land side of the barrier islands, showing larger differences between the "Baseline" and the "Ocean" scenario in the post-storm period. Note the plots have different y-axis ranges.

**References**

Huang, W., Zhang, Y.J., Wang, Z., Ye, F., Moghimi, S., Myers, E., Peeri, S., Yu, H.C.: Tide simulation revisited. Submitted to Ocean Modeling in April, 2021.

**End of Reply to Reviewer #2**

**Reply to Reviewer #3**

<Reviewer's general comment>

To me the novel aspect of this work is having very much detailed representation of hydrology (via NWM based on WRF), river morphology and coastal ocean dynamics (via a 3D baroclinic model). The manuscript is well written and can be of interest of community, thus I suggest publication in NHESS, however after a major revision. The major issue to me is that, despite the efforts of authors to calibrate/validate their model, on the hydrologic fluvial side the model does not seem well calibrated, and the significant error in NWM prediction (both timing and magnitude of peak) can pose a significant error to the overall estimates. Also, not clear to me how the pluvial flooding processes (from direct rainfall) is treated here. More detailed comments below.

<Response to general comment>

We need to clarify a possible misunderstanding on our model, because some descriptions in the original manuscript (OM) may be misleading. NWM (based on WRF-Hydro) is only used as a boundary condition like HYCOM and ERA, which all have uncertainties but are the state-of-the-art. We are not the developers of NWM, so the quality of NWM prediction is out of our control. It is the best hydrologic products available to us for this regional-scale study; we are fully open to other hydrologic models. NWM segments inside the model domain are only used to guide mesh generation, because these segments correspond to thalwegs in the DEM.

The merit of this work is being an effort toward a fully coupled compound inland flood and ocean related surge model. Here "fully-coupled" means the model solves its included processes with the same set of governing equations (Santiago-Collazo et al., 2019). Specifically, we include enough watershed region in our model domain so that the interaction between pluvial/fluvial processes and estuarine/oceanic processes can be simulated directly, without the need for "coupling" or "linking" among different types of models.

In the watershed, the movement of water (including flood routing) is NOT based on a specific "routing method" (such as "cell-to-cell" or "source-to-sink") as found in a tradition hydrologic model. The 1D Saint Venant equations or any of its approximated forms are NOT used at all. The hydrodynamic core of SCHISM, which solves the 3D Reynolds-averaged Navier–Stokes equation and transport equation, controls all water movements over land, in the rivers, estuaries and the ocean. The overland flow can occur anywhere in the model domain; there is no pre-defined stream segments as found in traditional hydrologic models that the flow must follow. This poses great challenge on the model's robustness and efficiency.

We removed the word "routing" throughout the manuscript.

We replaced the sentence on Line 134 of the OM:

"Watershed region of the grid has explicitly incorporated close to 300K NWM segments ("thalwegs"; cf. Fig. 3c), to facilitate the routing of river flow and precipitated water there."

with (Line 144, RM)

"Moreover, to better capture the geometry/bathymetry of potential flood pathways in the watershed region, specifically the region between the 10-m contour (set as the land boundary) and the 0-m contour of the DEM (based on NAVD88), about 300K National Water Model (NWM) segments (i.e., thalwegs; Fig. 3c) are reproduced in SCHISM's horizontal grid. Note that only the geometry of the NWM segments is retained,

while NWM outputs are only used as the land boundary condition and NWM does not solve any hydrodynamics inside the model domain."

We also added more details in "3.2 Coupling with NWM" (see responses to Reviewer's Major Comment #1 and Reviewer's Major Comment #2 below) to convey a clearer message on NWM's role in our model.

<Reviewer's Major Comment #1>

Hydrologic processes are not well represented here. River flow hydrograph estimates from NWM are significantly different than the observed flow at USGS gauges (see Figure 6). About 2-3 days of lag in peak flow estimation and up to 100% error in peak flow magnitude can pose a significant error in overall inundation and coastal water level dynamics if propagated through the system. Timing (and magnitude) of peak runoff plays a significant role in extreme water level dynamics in freshwater-influenced coastal systems and getting these hydrologic characteristics right plays a major role in accurate estimation of extent/intensity of compound flooding. This might partially explain the wide range of difference (up to 4 m) between simulated and estimated high water marks (Figure 11), with a considerable number of points having estimation error greater than 1m (Figure 11e).

<Response>

We agree with the reviewer that part of our model errors can be attributed to the errors from NWM. We add the following text after Line 188 of the original manuscript (Line 232, RM) to explain our choice:

"The forcing errors in the magnitude and timing of NWM's peak flow should explain part of the model errors especially in the watershed. For example, we found that replacing the NWM streamflow with the gaged flow at USGS Station 02109500 (Waccamaw River at Freeland, NC) improves the model skill locally. However, this is not cost-effective for our goal of operationalizing this compound flood model along the US East Coast and Gulf Coast. The developers of NWM (Gochis et al., 2018) showed that NWM's model skill was improved by each version update, with 44% of the gauges having bias < 20% in the latest version (NWM v2.0). We will adopt the newest and best NWM version in our ongoing study and operational forecast as soon as it is available. And we are open to using any other hydrologic sources to drive our model."

<Reviewer's Major Comment #2>

Could not find any information on how the pluvial processes are handled in the model. Rain-on grid modeling (L174) is a good contribution to the field which I acknowledge, but how the underlying processes and variables that contribute to the runoff generation are accounted for is vague. How infiltration capacity is accounted for? How drainage infrastructure contribution is accounted for (in urban settings)? How the generated runoff is routed between pixels? This is even more important when the results suggest that flooding in a significant portion of coastal land in dominated by "precipitation only"(Figure 15c).

<Response>

In the revised manuscript, we make it clear that all flow movements are solved by the same set of governing equations, i.e., Reynolds-averaged Navier–Stokes equations in hydrostatic form and transport equation.

And we clearly state that the neglection of infiltration and drainage is a limitation of the model and discuss the implications. We removed the original text on Line 171-173. The relevant texts in the revised manuscript read:

[Line 106, OM; Line 114, RM] This model solves the physical processes from the watershed to the ocean with the same set of governing equations, qualifying for Santiago-Collazo et al. (2019)'s definition of a fully-coupled compound surge and flood model. SCHISM (schism.wiki) uses efficient semi-implicit solvers to solve the hydrostatic form of the Reynolds-averaged Navier-Stokes equations and transport equation (Zhang et al., 2016), which govern all flow movements (including overland flow in the watersheds as well as estuarine and ocean circulations) inside the 3D model domain.

...

[Line 171-174, OM; Line 197, RM] Inside the model domain, streamflow, overland flow and precipitation are directly handled by the hydrodynamic core of SCHISM. This fully coupled configuration is rare in the existing compound flooding simulations (Santiago-Collazo et al., 2019). To ensure the accuracy and robustness of SCHISM in simulating hydrological and hydraulic processes including the overland flow, we already examined the model's performance in both lab-scale and field-scale tests in a previous study (Section 2.2 and 2.3 of Zhang et al. (2020)) and applied the model in the Delaware Bay watershed including the Delaware River with a hydraulic jump up to 40 meters above the NAVD88 datum (Fig. 14 in Zhang et al. (2020)). The NWM segments explicitly reproduced in our grid (Fig. 3c) during the mesh generation stage help capture the bathymetry of main flood pathways (thalwegs). However, flow is not restricted to these 1D segments"

…

[Line 184, OM; Line 215, RM] As a model limitation, infiltration is neglected in this work. In the case of Hurricane Florence induced flooding, we expect the effect of infiltration to be minor. According to NOAA's weather map (https://www.wpc.ncep.noaa.gov/dailywxmap/), there was continuous rainfall along the US east coast from Sep 11, 2018 to the date of Florence's landfall (Sep 14, 2018), so the infiltration capacity of the soil was already reduced. Moreover, "wet" storms like Hurricane Florence (2018) and Hurricane Harvey (2017) tend to dump large amount of rain fall at a location for days because of the slow movement of the storm, so most of the rainfall is on saturated soil. As another model limitation, the drainage in urban settings is not included in our model. This may lead to some occasional big errors in the prediction of elevations on high water marks for example the one large error in the urban area in Fig. 13d. We do have a plan of explicitly accounting for infiltration as volume sinks based on NWM (or other hydrologic models). However, considering the additional uncertainty this would bring, for now we choose to continue improving more important aspects of the model (especially the quality of model grid, which is likely responsible for most of the large errors in Fig. 11c) for operational use.

…

[Line 300-302, OM; Line 364, RM] The only remaining large error point in the "baseline" occurs in an urban area away from the river, likely due to the building or drainage effects that have not been incorporated in the model (Fig. 13d).

…

[Beginning of "Section 6: Conclusions"] We have successfully applied a 3D cross-scale model to examine the compound flooding processes that occurred during Hurricane Florence. The model is fully coupled in the sense that the hydrologic and hydrodynamic processes are solved by the same set of governing equations.

Limitations of the model include the neglection of infiltration and urban drainage, which will be implemented soon.

< Reviewer's Minor Comment #1>

Figure 1: red text (in the left panels) is not readable in print. I'd also add some space between rows in the legend.

<Response>

Revised as suggested:

[Figure]

<Reviewer's Minor Comment #2>

Figure 3: Black text is really hard to read on the dark blue background.

<Response>

Revised as suggested:

[Figure]

Fig. 3: Model domain and horizontal grid. (a) Domain extent and hurricane track. (b) Station locations along the North Carolina and South Carolina coast. The six NOAA gauges are: Charleston (8665530); Springmaid Pier (8661070); Wrightsville Beach (8658163); Beaufort (8656483); Hatteras (8654467); Oregon Inlet (8652587). The three squares are NDBC buoys (41013, 41159, 41025). The spatial extents of (c), (d), and (f) are also marked in (b). (c) Zoom-in of grid in a watershed area (the arcs are from NWM river network). (d) Zoom-in of grid near barrier islands and inlet (the dark line is the 0 m isobath, NAVD88); (e) Cumulative histogram of grid resolution (measured in equivalent diameters); (f) Grid resolution in North Carolina's coastal watershed area.

<Reviewer's Minor Comment #3>

Figure 4 is confusing. Hurricane track is numbered right-to-left (Panel a) and vertical grid points are numbered Left-to-Right (Panels b and c)

<Response>

Revised as suggested. The numbered points are removed and the transect is more clearly defined with along-transect distance marked in all subplots.

[Figure]

Fig. 4: Vertical grid. (a) Transect from the watershed to the ocean, used to illustrate the vertical grid; (b) vertical grid along the transect; (c) zoom-in from (b), illustrating the transitions from 3D (Pamlico Sound) to 2DH (barrier island) and back to 3D (coastal ocean).

<Reviewer's Minor Comment #4>

L110: what "LSC" stands for?

<Response>

We change this sentence to (Line 120, RM):

"… that combines a hybrid triangular-quadrangular unstructured grid in the horizontal dimension and localized sigma coordinates with shaved cells (dubbed as LSC2; Zhang et al., 2015) in the vertical dimension."

<Reviewer's Minor Comment #5>

L120: "10 m above sea level", you mean mean sea level? if yes, is it regional or global. Please be more specific about this.

<Response>

(Line 79, RM) Changed to "10 m above the NAVD88 datum".

<Reviewer's Minor Comment #6>

L133: what "CFL" stands for?

<Response>

(Line 143, RM) Changed to "Courant–Friedrichs–Lewy (CFL) condition"

<Reviewer's Minor Comment #7>

L143: Explain what is HYCOM first time use it in the manuscript.

<Response>

(Line 157, RM) We use the term "HYbrid Coordinate Ocean Model (HYCOM; https://www.hycom.org/)" upon the first occurrence.

<Reviewer's Minor Comment #8>

L174: Simply said rainfall is routed. Please, elaborate more on this, how exactly routed? How infiltration capacity incorporated?

Please see the response to <Reviewer's Major Comment #2>.

<Reviewer's Minor Comment #9>

L189: "4-6 days" in observed or modeled events? The modeled and simulated peaks seem to be few days apart them-selves.

<Response>

In the revised manuscript, we clearly labeled the date of the peaks in Fig. 6:

[Figure]

(Line 230, RM) We change the sentence to:

"The observation indicates the peak streamflow occurs about 7 days after the landfall, which is the time it takes for the rainfall induced flood to reach the coastal rivers. Note that there is typically a time lag of 1-2

days between the peak flow in NWM and the gaged flow (Fig. 6). The forcing errors in the magnitude and timing of NWM's peak flow should explain part of the model errors especially in the watershed."

<Reviewer's Minor Comment #10>

Figure 9: Better to compare detided signal here. These gauges are extremely tidal influenced and estimation skills of the model for stochastic processes is diluted by highly predictable astronomic tides.

<Response>

We agree that de-tided signal will give a clearer picture of stochastic processes. We add new sub-plots on sub-tidal signals (see the figure below) to Fig. 9.

[Figure]

Fig. 9: Comparison of elevation at six NOAA gauges: (left) total elevation; (right) subtidal elevation. Also included are results from two sensitivity runs ("Baseline_Wave" and "No_NWM_precipOcean"; cf. Table 1). The stations in (a)-(d) are exposed to the ocean, showing small differences among scenarios; the two stations in (e) and (f) are on the land side of the barrier islands, showing larger differences between the "Baseline" and the "Ocean" scenario in the post-storm period. Note the plots have different y-axis ranges.

We also add more discussion in the text:

(Line 242, OM; Line 292, RM) "…, as seen from the total elevation (Fig. 9a) and the sub-tidal signals (Fig. 9b). The latter applies a low-pass Butterworth filter (Butterworth, 1930) only preserving longer-period (longer than 2 days) components."

(Line 251, OM; Line 302, RM) "The mechanism causing the water level set-downs at the two South Carolina stations is similar to that causing the set-downs behind the barrier islands in North Carolina. The two South Carolina stations (Charleston and Springmaid) are located to the south of the landfall site and the wind direction is from the land to the ocean, pushing water away from shore."

(Line 253, OM; Line 306, RM) "The averaged MAE for the subtidal comparison is 8.6 cm and averaged correlation coefficient is 0.92."

<Reviewer's Minor Comment #11>

L275 & L284: describing errors "quite satisfactory" and "reasonable agreement", when mean absolute error in estimated high water levels is 0.73 m and with nearly 20% of points having error >1m, is a subjective description. I'd avoid such statements and simply let the readers decide whether if such estimation errors are reasonable according to their required accuracy and project objectives.

<Response>

As suggested, we have removed all subjective claims such as "quite satisfactory" and "reasonable agreement".

**<References>**

Butterworth, S. (1930), On the theory of filter amplifiers, Wireless Eng., 7, 536–541.

Gochis, D.J., Cosgrove, B., Dugger, A.L., Karsten, L., Sampson, K.M., McCreight, J.L., Flowers, T., Clark, E.P., Vukicevic, T., Salas, F. and FitzGerald, K., 2018, December. Multi-variate evaluation of the NOAA National Water Model. In AGU Fall Meeting Abstracts (Vol. 2018, pp. H33G-01).

Santiago-Collazo, F.L., Bilskie, M.V., Hagen, S.C.: A comprehensive review of compound inundation models in low-gradient coastal watersheds. Environ. Model. Software 119:166–181. https://doi.org/10.1016/j.envsoft.2019.06.002, 2019.

Zhang, Y.J., Ye, F., Stanev, E.V. and Grashorn, S., 2016. Seamless cross-scale modeling with SCHISM. Ocean Modelling, 102, pp.64-81.

Zhang, Y.J., Ye, F., Yu, H., Sun, W., Moghimi, S., Myers, E., Nunez, K., Zhang, R., Wang, H., Roland, A. and Du, J., 2020. Simulating compound flooding events in a hurricane. Ocean Dynamics, 70(5), pp.621-640.

**<End of Reply to Reviewer #3>**

**Additional changes**

We found a minor problem with Fig. 13d and fixed it in the new Fig. 13 (shown below) and Fig. 1d. Specifically, a different run other than the baseline model was used to plot Fig. 13d in the original manuscript, which has been corrected. This does not change the conclusion that resolving small-scale flow routing features greatly improves the prediction of HWM elevation. The "Burnt Mill Creek" was mis-labeled as "Northeast Cape Fear River" in the original manuscript, which has been corrected. Lake Greenfield and more labels were also added for better illustration.

[revised manuscript text omitted]

---

## Author Response (AR2)

**Authors' Reply**

Dear reviewers and editor,

Thank you for your time and effort in helping us improve the manuscript.

We address the two remaining minor comments here:

<Comment #1>

Please clarify this following statement: "but the sensitivity is small in the range of 0.001-0.005". What exactly did the authors mean by sensitivity is small? Sensitivity of what (error?) to what (friction factors?)? Do the values of 0.001-0.01, 0.001-0.005 have a unit?

<Response>

The reviewer's interpretation is correct. We have changed the sentence to:

"In the wet area, drag coefficients within a range of 0.001-0.01 (non-dimensional) were tested. Commonly accepted default value 0.0025 gave good error statistics near the landfall site and values within a range of 0.001-0.005 gave very similar results."

<Comment #2>

In terms of the return period, I was referring to the return period of water levels, which can be estimated using historical observations if they exist at selected locations. But I am happy with the authors' proposed alternative solution.

<Response>

Thanks. We choose to keep the manuscript as is, but we will keep this in mind in our future study.

In addition, we made some minor changes only in the text, which are highlighted in the "track-change" version of the manuscript.